# On the Clean Generalization and Robust Overfitting in Adversarial Training from Two Theoretical Views: Representation Complexity and Training Dynamics

**Binghui Li** [1]    **Yuanzhi Li** [2]

## Abstract

Similar to surprising performance in the standard deep learning, deep nets trained by adversarial training also generalize well for *unseen clean data (natural data)*. However, despite adversarial training can achieve low robust training error, there exists a significant *robust generalization gap*. We call this phenomenon the *Clean Generalization and Robust Overfitting (CGRO)*. In this work, we study the CGRO phenomenon in adversarial training from two views: *representation complexity* and *training dynamics*. Specifically, we consider a binary classification setting with $N$ separated training data points. *First*, we prove that, based on the assumption that we assume there is $\text{poly}(D)$-size clean classifier (where $D$ is the data dimension), ReLU net with only $\tilde{O}(ND)$ extra parameters is able to leverages robust memorization to achieve the CGRO, while robust classifier still requires exponential representation complexity in worst case. *Next*, we focus on a structured-data case to analyze training dynamics, where we train a two-layer convolutional network with $\tilde{O}(ND)$ width against adversarial perturbation. We then show that a three-stage phase transition occurs during learning process and the network provably converges to robust memorization regime, which thereby results in the CGRO. *Besides*, we also empirically verify our theoretical analysis by experiments in real-image recognition datasets.

## 1. Introduction

Nowadays, deep neural networks have achieved excellent performance in a variety of disciplines, especially including in computer vision (Krizhevsky et al., 2012; Dosovitskiy et al., 2020; Kirillov et al., 2023) and natural language process (Devlin et al., 2018; Brown et al., 2020; Ouyang et al., 2022). However, it is well-known that small but adversarial perturbations to the natural data can make well-trained classifiers confused (Biggio et al., 2013; Szegedy et al., 2013; Goodfellow et al., 2014), which potentially gives rise to reliability and security problems in real-world applications and promotes designing adversarial robust learning algorithms.

In practice, adversarial training methods (Goodfellow et al., 2014; Madry et al., 2017; Shafahi et al., 2019; Zhang et al., 2019; Pang et al., 2022) are widely used to improve the robustness of models by regarding perturbed data as training data. However, while these robust learning algorithms are able to achieve high robust training accuracy (Gao et al., 2019), it still leads to a non-negligible robust generalization gap (Raghunathan et al., 2019), which is also called *robust overfitting* (Rice et al., 2020; Yu et al., 2022).

To explain this puzzling phenomenon, a series of works have attempted to provide theoretical understandings from different perspectives. Despite these aforementioned works seem to provide a series of convincing evidence from theoretical views in different settings, there still exists *a gap between theory and practice* for at least *two reasons*.

*First*, although previous works have shown that adversarial robust generalization requires more data and larger models (Schmidt et al., 2018; Gowal et al., 2021; Li et al., 2022; Bubeck & Sellke, 2023), it is unclear that what mechanism, during adversarial training process, *directly* causes robust overfitting. A line of work about uniform algorithmic stability (Xing et al., 2021; Xiao et al., 2022), under Lipschitzian smoothness assumptions, also suggest that robust generalization gap increases when training iteration is large. In other words, we know there is no robust generalization gap for a trivial model that only guesses labels totally randomly (e.g. deep neural networks at random initialization), which implies that we should take learning process into consideration to analyze robust generalization.

*Second and most importantly*, while some works (Tsipras et al., 2018; Zhang et al., 2019; Hassani & Javanmard, 2022) point out that achieving robustness may hurt clean test accuracy, in most of the cases, it is observed that drop of robust

---

[1] Center for Machine Learning Research, Peking University [2] Machine Learning Department, Carnegie Mellon University. Correspondence to: Binghui Li <libinghui@pku.edu.cn>, Yuanzhi Li <yuanzhil@andrew.cmu.edu>.

*Proceedings of the $42^{nd}$ International Conference on Machine Learning*, Vancouver, Canada. PMLR 267, 2025. Copyright 2025 by the author(s).

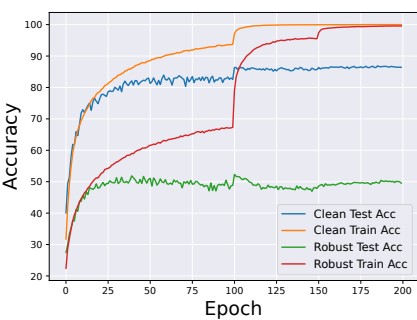

*Figure 1.* The learning curves of adversarial training on CIFAR10 with $\ell_\infty$-perturbation radius $\delta = 8/255$ (Rice et al., 2020).

test accuracy is much higher than drop of clean test accuracy in adversarial training (Madry et al., 2017; Schmidt et al., 2018; Raghunathan et al., 2019) (see in Figure 1, where clean test accuracy is more than $80\%$ but robust test accuracy only attains nearly $50\%$). Namely, a weak version of benign overfitting (Zhang et al., 2017), which means that overparameterized deep neural networks can both fit random data powerfully and generalize well for unseen *clean* data, remains after adversarial training. Therefore, it is natural to ask the following question:

> *What is the underlying mechanism that results in both Clean Generalization and Robust Overfitting (CGRO) during adversarial training?*

In this paper, we provide a theoretical understanding of this question. Precisely, we make the following contributions:

- In Section 3, we first present some useful notations used in our work, and then provide the formal definition of the CGRO classifier (Definition 3.4).

- In Section 4, we study the CGRO classifiers via the view of representation complexity. Based on some data assumptions observed in practice, we prove that achieving CGRO classifier only needs extra *linear* parameters by leveraging robust memorization (Theorem 4.4), but robust classifier requires even *exponential* model capacity in worst case (Theorem 4.7).

- In Section 5, under our theoretical framework of adversarial training, we apply a two-layer convolutional network to learn the structured data. We propose a *three-stage* analysis technique to decouple the complicated training dynamics of adversarial training, which shows that the network learner provably converges to the CGRO regime (Theorem 5.9).

- In Section 6, we empirically demonstrate our theoretical results in Section 4 and Section 5 by experiments in real-world data and synthetic data, respectively.

## 2. Additional Related Work

**Empirical Works on Robust Overfitting.** One surprising behavior of deep learning is that over-parameterized neural networks can generalize well, which is also called *benign overfitting* that deep models have not only the powerful memorization but a good performance for unseen data (Zhang et al., 2017; Belkin et al., 2019). However, in contrast to the standard (non-robust) generalization, for the robust setting, Rice et al. (2020) empirically investigates robust performance of models based on adversarial training methods, which are used to improve adversarial robustness (Szegedy et al., 2013; Madry et al., 2017), and the work Rice et al. (2020) shows that *robust overfitting* can be observed on multiple datasets, including CIFAR10 and ImageNet.

**Theoretical Works on Robust Overfitting.** A list of works (Schmidt et al., 2018; Balaji et al., 2019; Dan et al., 2020) study the *sample complexity* for adversarial robustness, and their works manifest that adversarial robust generalization requires more data than the standard setting, which gives an explanation of the robust generalization gap from the perspective of statistical learning theory. And another line of works (Tsipras et al., 2018; Zhang et al., 2019) propose a principle called the *robustness-accuracy trade-off* and have theoretically proven the principle in different setting, which mainly explains the widely observed drop of robust test accuracy due to the trade-off between adversarial robustness and clean accuracy. Recently, Li et al. (2022) investigates the robust expressive ability of neural networks and shows that robust generalization requires exponentially large models.

**Feature Learning Theory of Deep Learning.** The *feature learning theory* of neural networks (Allen-Zhu & Li, 2020a;b; 2022; Shen et al., 2022; Jelassi & Li, 2022; Jelassi et al., 2022; Chen et al., 2022; Chidambaram et al., 2023) is proposed to study how features are learned in deep learning tasks, which provide a theoretical analysis paradigm beyond the *neural tangent kernel (NTK) theory* (Jacot et al., 2018; Du et al., 2018; 2019; Allen-Zhu et al., 2019; Arora et al., 2019). In this work, we make a first step to understand clean generalization and robust overfitting (CGRO) phenomenon in adversarial training by analyzing feature learning process under our theoretical framework about structured data.

**Memorization in Adversarial Training.** Dong et al. (2021); Xu et al. (2021) empirically and theoretically explore the memorization effect in adversarial training for promoting a deeper understanding of model capacity, convergence, generalization, and especially robust overfitting of the adversarially trained models. However, different from their works, the concept *clean generalization and robust overfitting (CGRO)* proposed in our paper focuses on both robust overfitting and high clean test accuracy, which means that there is surprisingly no clean memorization or clean overfitting.

## 3. Preliminaries

In this section, we first introduce some useful notations that are used in this paper, and then present our problem setup.

### 3.1. Notations

Throughout this work, we use letters for scalars and bold letters for vectors. We will use $[k]$ to indicate the index set $\{1, 2, \cdots, k\}$. The indicator of an event is defined as $\mathbb{I}\{\cdot\} = 1$ if the event $\cdot$ holds and $\mathbb{I}\{\cdot\} = 0$ otherwise. We use $\mathrm{sgn}(\cdot)$ to denote the sign function of the real number $\cdot$, and we use $\mathrm{span}(\boldsymbol{v}_1, \boldsymbol{v}_2, \cdots, \boldsymbol{v}_n)$ to denote the linear span of the vectors $\boldsymbol{v}_1, \boldsymbol{v}_2, \cdots, \boldsymbol{v}_n \in \mathbb{R}^D$.

For any given two sequences $\{A_n\}_{n=0}^{\infty}$ and $\{B_n\}_{n=0}^{\infty}$, we denote $A_n = O(B_n)$ if there exist some absolute constant $C_1 > 0$ and $N_1 > 0$ such that $|A_n| \leq C_1 |B_n|$ for all $n \geq N_1$. Similarly, we denote $A_n = \Omega(B_n)$ if there exist $C_2 > 0$ and $N_2 > 0$ such that $|A_n| \geq C_2 |B_n|$ for all $n > N_2$. We say $A_n = \Theta(B_n)$ if $A_n = O(B_n)$ and $A_n = \Omega(B_n)$ both holds. We use $\widetilde{O}(\cdot), \widetilde{\Omega}(\cdot)$, and $\widetilde{\Theta}(\cdot)$ to hide logarithmic factors in these notations respectively. Moreover, we denote $A_n = \mathrm{poly}(B_n)$ if $A_n = O(B_n^K)$ for some positive constant $K$, and $A_n = \mathrm{polylog}(B_n)$ if $B_n = \mathrm{poly}(\log(B_n))$. We also say $A_n = o(B_n)$ if for arbitrary positive constant $C_3 > 0$, there exists $N_3 > 0$ such that $|A_n| < C_3 |B_n|$ for all $n > N_3$. An event is said that it happens with high probability (or w.h.p. for short) if it happens with probability at least $1 - o(1)$.

We use the notation $\|\cdot\|_p, p \in [1, +\infty]$ to denote the $\ell_p$ norm in the vector space $\mathbb{R}^D$. For two sets $\mathcal{A}, \mathcal{B} \subset \mathbb{R}^D$, we can define the $\ell_p$-distance between $\mathcal{A}$ and $\mathcal{B}$ as $\mathrm{dist}_p(\mathcal{A}, \mathcal{B}) := \inf\{\|\boldsymbol{X} - \boldsymbol{Y}\|_p : \boldsymbol{X} \in \mathcal{A}, \boldsymbol{Y} \in \mathcal{B}\}$. For $r > 0$, $\mathbb{B}_p(\boldsymbol{X}, r) := \left\{\boldsymbol{Z} \in \mathbb{R}^D : \|\boldsymbol{Z} - \boldsymbol{X}\|_p \leq r\right\}$ is defined as the $\ell_p$-ball with radius $r$ centered at $\boldsymbol{X}$.

A multi-layer neural network is a function from input in $\mathbb{R}^D$ to output in $\mathbb{R}^m$, which is defined as follows:

$$\boldsymbol{F}_{\boldsymbol{W}, \boldsymbol{B}}(\boldsymbol{X}) := \boldsymbol{W}_L \sigma \left(\boldsymbol{W}_{L-1} \sigma \left(\cdots \boldsymbol{W}_1 \sigma \left(\boldsymbol{W}_0 \boldsymbol{X} + \boldsymbol{B}_0\right) + \boldsymbol{B}_1 \cdots\right) + \boldsymbol{B}_{L-1}\right) + \boldsymbol{B}_L$$

and

$$\boldsymbol{W}_0 \in \mathbb{R}^{m_0 \times D}, \quad \boldsymbol{W}_i \in \mathbb{R}^{m_i \times m_{i-1}}, 1 \leq i \leq L-1,$$
$$\boldsymbol{W}_L \in \mathbb{R}^{m \times m_{L-1}}, \quad \boldsymbol{B}_i \in \mathbb{R}^{m_i}, 0 \leq i \leq L-1,$$
$$\boldsymbol{B}_L \in \mathbb{R}^m, \quad \boldsymbol{X} \in \mathbb{R}^D, \quad \boldsymbol{F}_{\boldsymbol{W}, \boldsymbol{B}}(\boldsymbol{X}) \in \mathbb{R}^m,$$

where the weight $\boldsymbol{W} = [\boldsymbol{W}_0, \boldsymbol{W}_1, \cdots, \boldsymbol{W}_L]$ and the bias $\boldsymbol{B} = [\boldsymbol{B}_0, \boldsymbol{B}_1, \cdots, \boldsymbol{B}_L]$ are learnable parameters, and $\max\{D, m_0, m_1, \cdots, m_{L-1}, m\}$ and $L+1$ are the width and depth of the neural network, respectively. We use $\sigma(\cdot)$ to denote the (non-linear) entry-wise activation function, and ReLU activation function is defined as $\sigma(\cdot) := \max(\cdot, 0)$.

### 3.2. Problem Setup

We consider a binary classification setting, where we use $\boldsymbol{X} \in \mathcal{X} \subset \mathbb{R}^D$ to denote the data input and the binary label $y$ is in $\mathcal{Y} = \{-1, 1\}$. Given a data distribution $\mathcal{D}$ that is a joint distribution over the supporting set $\mathcal{X} \times \mathcal{Y}$, and a function $f : \mathcal{X} \to \mathbb{R}$ as the classifier, we can define the following measurements to describe the clean (robust) classification performance of the classifier $f$ on data $\mathcal{D}$.

**Definition 3.1.** (Clean Test Error) The clean test error of the classifier $f$ w.r.t. the data distribution $\mathcal{D}$ is defined as $\mathcal{L}_{\mathcal{D}}(f) := \mathbb{P}_{(\boldsymbol{X}, y) \sim \mathcal{D}}[\mathrm{sgn}(f(\boldsymbol{X})) \neq y]$.

**Definition 3.2.** (Robust Test Error) Given a $\ell_p$-robust radius $\delta \geq 0$, the robust test error of the classifier $f$ w.r.t. the data distribution $\mathcal{D}$ and $\delta$ under $\ell_p$ norm is defined as $\mathcal{L}_{\mathcal{D}}^{p, \delta}(f) := \mathbb{E}_{(\boldsymbol{X}, y) \sim \mathcal{D}}\left[\max_{\|\boldsymbol{X'} - \boldsymbol{X}\|_p \leq \delta} \mathbb{I}\{\mathrm{sgn}(f(\boldsymbol{X'})) \neq y\}\right]$.

In our work, we mainly focus on the cases when $p = 2$ and $p = \infty$, which can be extended to the general $p$ case as well.

In adversarial training, with access to the training dataset $\mathcal{S} = \{(\boldsymbol{X}_1, y_1), (\boldsymbol{X}_2, y_2), \ldots, (\boldsymbol{X}_N, y_N)\}$ randomly sampled from the data distribution $\mathcal{D}$, we aim to minimize the following robust training error to derive the robust classifier.

**Definition 3.3.** (Robust Training Error) Given a $\ell_p$-robust radius $\delta \geq 0$, the robust training error of the classifier $f$ w.r.t. training dataset $\mathcal{S}$ and $\delta$ under $\ell_p$ norm is defined as $\mathcal{L}_{\mathcal{S}}^{p, \delta}(f) := \frac{1}{N} \sum_{i=1}^{N} \max_{\|\boldsymbol{X}_i' - \boldsymbol{X}_i\|_p \leq \delta} \mathbb{I}\{\mathrm{sgn}(f(\boldsymbol{X}_i')) \neq y\}$.

Now, we present the concept **CGRO classifier** that we mainly study in this paper as the following definition.

**Definition 3.4.** (CGRO Classifier) Given a $\ell_p$-robust radius $\delta \geq 0$, we say a classifier $f$ is CGRO classifier w.r.t. the data distribution $\mathcal{D}$ and training dataset $\mathcal{S}$ if it satisfies that $\mathcal{L}_{\mathcal{D}}(f) = o(1)$, $\mathcal{L}_{\mathcal{S}}^{p, \delta}(f) = o(1)$ but $\mathcal{L}_{\mathcal{D}}^{p, \delta}(f) = \Omega(1)$.

**Remark 3.5.** *In the above definition of CGRO classifier, the asymptotic notations $o(1)$ and $\Omega(1)$ are both defined with respective to the data dimension $D$, which means that CGRO classifier has a good clean test performance but a poor robust generalization at the same time.*

## 4. Analyzing the CGRO Phenomenon from the View of Representation Complexity

In this section, we provide a theoretical understanding of the CGRO phenomenon from the view of representation complexity. First, We present the data assumption as follow.

For the data distribution $\mathcal{D}$ and $\ell_p$-robust radius $\delta > 0$, the supporting set of the data input $\mathcal{X}$ can be divided into two sets $\mathcal{X}_+$ and $\mathcal{X}_-$ that correspond to the positive and negative classes respectively, i.e. $\mathcal{X}_+ := \{\boldsymbol{X} \in \mathcal{X} : (\boldsymbol{X}, y) \in \mathcal{D}, y = 1\}$ and $\mathcal{X}_- := \{\boldsymbol{X} \in \mathcal{X} : (\boldsymbol{X}, y) \in \mathcal{D}, y = -1\}$.

**Assumption 4.1.** (Bounded) There exists a absolute constant $\mathcal{R} > 0$ such that, with high probability over the data distribution $\mathcal{D}$, it holds that the data input $\boldsymbol{X} \in [-\mathcal{R}, \mathcal{R}]^D$.

Recall the definition of CGRO classifier (Definition 3.4). W.l.o.g., we can only focus on the case $\mathcal{X} \subset [-\mathcal{R}, \mathcal{R}]^D$.

**Assumption 4.2.** (Well-Separated) we assume that the separation between the positive and negative classes is large than twice the robust radius, i.e. $\text{dist}_p(\mathcal{X}_+, \mathcal{X}_-) > 2\delta$.

This assumption is necessary to ensure the existence of a robust classifier, and which is indeed empirically observed in real-world image classification data (Yang et al., 2020).

**Assumption 4.3.** (Polynomial-Size Clean Classifier Exists) we assume that there exists a clean classifier $f_{clean} : \mathcal{X} \to \mathbb{R}$ represented as a ReLU network with $\text{poly}(D)$ parameters such that $\mathcal{L}_{\mathcal{D}}(f_{clean}) = o(1)$ but $\mathcal{L}_{\mathcal{D}}^{p,\delta}(f_{clean}) = \Omega(1)$.

In Assumption 4.3, the asymptotic notations $o(1)$ and $\Omega(1)$ are defined w.r.t. the data dimension $D$. It means that the clean classifier with moderate size can perfectly classify the natural data but fails to classify the adversarially perturbed data, which is consistent with the common practice that well-trained neural networks are vulnerable to adversarial examples (Szegedy et al., 2013; Raghunathan et al., 2019).

Now, we present our main result about the upper bound of representation complexity for CGRO classifier as follow.

**Theorem 4.4.** (Polynomial Upper Bound for CGRO) Under Assumption 4.1, 4.2 and 4.3, with $N$-sample training dataset $\mathcal{S} = \{(\boldsymbol{X}_1, y_1), (\boldsymbol{X}_2, y_2), \ldots, (\boldsymbol{X}_N, y_N)\}$ drawn from the data distribution $\mathcal{D}$, there exists a CGRO classifier $f_{CGRO} : \mathcal{X} \to \mathbb{R}$ that can be represented as a ReLU network with at most $\text{poly}(D) + \tilde{O}(ND)$ parameters.

**Proof Sketch.** First, we show the existence of the CGRO classifier w.r.t the data distribution $\mathcal{D}$ and training dataset $\mathcal{S}$.

**Lemma 4.5.** For given the data distribution $\mathcal{D}$ satisfying Assumption 4.1, 4.2 and 4.3, and the randomly sampled training dataset $\mathcal{S} = \{(\boldsymbol{X}_1, y_1), (\boldsymbol{X}_2, y_2), \ldots, (\boldsymbol{X}_N, y_N)\}$, we consider the following function $f_{\mathcal{S}} : \mathcal{X} \to \mathbb{R}$ defined as

$$f_{\mathcal{S}}(\boldsymbol{X}) = \underbrace{f_{clean}(\boldsymbol{X}) \left(1 - \mathbb{I}\{\boldsymbol{X} \in \cup_{i=1}^{N} \mathbb{B}_p(\boldsymbol{X}_i, \delta)\}\right)}_{\text{clean classification on unseen test data}}$$
$$+ \underbrace{\sum_{i=1}^{N} y_i \mathbb{I}\{\boldsymbol{X} \in \mathbb{B}_p(\boldsymbol{X}_i, \delta)\}}_{\text{robust memorization on training data}}.$$

And it holds that the function $f_{\mathcal{S}}$ is a CGRO classifier.

Due to Assumption 4.2, it is clear that we have $\mathbb{B}_p(\boldsymbol{X}_i, \delta) \cap \mathbb{B}_p(\boldsymbol{X}_j, \delta) = \emptyset$ for any distinct $i, j \in [N]$. Thus, $f_{\mathcal{S}}$ perfectly clean classifies unseen test data by the first summand

(Assumption 4.3) and robustly memorizes training data by the second summand, which belongs to CGRO classifiers.

Back to the proof of Theorem 4.4, the key idea is to approximate the classifier $f_{\mathcal{S}}$ proposed in Lemma 4.5 by ReLU network. Indeed, the function $f_{\mathcal{S}}$ can be rewritten as

$$f_{\mathcal{S}}(\boldsymbol{X}) = \underbrace{\sum_{i=1}^{N} (y_i - f_{clean}(\boldsymbol{X})) \mathbb{I}\{\|\boldsymbol{X} - \boldsymbol{X}_i\|_p \le \delta\}}_{\text{weighted sum of robust local indicators}}$$
$$+ \underbrace{f_{clean}(\boldsymbol{X})}_{\text{poly}(D)}.$$

Therefore, we use ReLU nets to approximate the distance function $d_i(\boldsymbol{X}) := \|\boldsymbol{X} - \boldsymbol{X}_i\|_p$ in $[-\mathcal{R}, \mathcal{R}]^D$ efficiently, and we also noticed that the indicator $\mathbb{I}\{\cdot\}$ can be approximated by a soft indicator represented by two ReLU neurons. Combined with the above results, there exists a ReLU net $f_{CGRO}$ with $\text{poly}(D) + \tilde{O}(ND)$ parameters such that $\|f_{CGRO} - f_{\mathcal{S}}\|_\infty = o(1)$, which implies Theorem 4.4. $\square$

**Remark 4.6.** *Theorem 4.4 manifests that ReLU net with only $\tilde{O}(ND)$ extra parameters is able to leverages robust memorization (Lemma 4.5) to achieve the CGRO regime.*

However, to achieve robust generalization, higher complexity seems necessary. We generalize the result in Li et al. (2022) from linear-separable setting to Assumption 4.3.

**Theorem 4.7.** (Exponential Lower Bound for Robust Classifer) Let $\mathcal{F}_M$ be the family of function represented by ReLU networks with at most $M$ parameters. There exists a number $M_D = \Omega(\exp(D))$ and a distribution $\mathcal{D}$ satisfying Assumption 4.1, 4.2 and 4.3 such that, for any classifier in the family $\mathcal{F}_{M_D}$, the robust test error w.r.t. $\mathcal{D}$ is at least $\Omega(1)$.

**Proof Sketch.** The main proof idea of Theorem 4.7 is the linear region decomposition of ReLU nets (Montufar et al., 2014), and then we apply the technique bounding the VC dimension for local region similar to Li et al. (2022). $\square$

According to Assumption 4.3, Theorem 4.4 and Theorem 4.7, we obtain the following inequalities about representation complexity of classifiers in different regimes.

$$\underbrace{\textit{Clean Classifier}}_{\text{poly}(D)} \lesssim \underbrace{\textit{CGRO Classifier}}_{\text{poly}(D) + \tilde{O}(ND)} \ll \underbrace{\textit{Robust Classifier}}_{\Omega(\exp(D))}.$$

**Remark 4.8.** *These inequalities states that while CGRO classifiers have mildly higher representation complexity than clean classifiers, adversarial robustness requires excessively higher complexity, which may lead the classifier trained by adversarial training to the CGRO regime. We also empirically verify this by the experiments of adversarial training regarding different model sizes in Section 6.*

# 5. Analyzing Dynamics of Adversarial Training on Structured Data

In the previous section, we show the efficiency of CGRO classifier via representation complexity. However, it is unclear how the classifier trained by adversarial training converges to the CGRO regime. In this section, we will focus on a structured-data setting to study the learning process of adversarial training. First, we introduce our structured-data setting as follow, and then provide a fine-grained analysis of adversarial training under our theoretical framework.

## 5.1. Structured Data Setting

In this section, we consider the case that data input $\boldsymbol{X} \in \mathbb{R}^D$ has a patch structure as follow, which is similar to a list of theoretical works about feature learning (Allen-Zhu & Li, 2020b; Chen et al., 2021; Jelassi & Li, 2022; Jelassi et al., 2022; Kou et al., 2023; Chidambaram et al., 2023).

**Definition 5.1.** (Patch Data Distribution) We define a data distribution $\mathcal{D}$, in which each instance consists in an input $\boldsymbol{X} \in \mathcal{X} = \mathbb{R}^D$ and a label $y \in \mathcal{Y} = \{-1, 1\}$ generated by

1.  The label $y$ is uniformly drawn from $\{-1, 1\}$.

2.  The input $\boldsymbol{X} = (\boldsymbol{X}[1], \ldots, \boldsymbol{X}[P])$, where each patch $\boldsymbol{X}[j] \in \mathbb{R}^d$ and $P = D/d$ is the number of patches (we assume that $D/d$ is an integer and $P = \mathrm{polylog}(d)$).

3.  Meaningful Signal patch: for each instance, there exists one and only one meaningful patch $\mathrm{signal}(\boldsymbol{X}) \in [P]$ satisfies $\boldsymbol{X}[\mathrm{signal}(\boldsymbol{X})] = \alpha y \boldsymbol{w}^*$, where $\boldsymbol{w}^* \in \mathbb{R}^d(\|\boldsymbol{w}^*\|_2 = 1)$ is the unit meaningful signal vector and $\alpha > 0$ is the norm of signal.

4.  Noisy patches: $\boldsymbol{X}[j] \sim \mathcal{N}\left(0, \left(\mathbf{I}_d - \boldsymbol{w}^*\boldsymbol{w}^{*\top}\right)\sigma_p^2\right)$, for $j \in [P]\backslash\{\mathrm{signal}(\boldsymbol{X})\}$, where $\sigma_p^2 > 0$ denotes the variance of noise .

**Remark 5.2.** *The patch structure that we leverage can be viewed as a simplification of real-world vision-recognition datasets. Indeed, images are divided into signal patches that are meaningful for the classification such as the whisker of a cat or the nose of a dog, and noisy patches like the uninformative background of a photo. And our patch data assumption can also be generalized to the case that there exist a set of patches that are meaningful, while analyzing the learning process will be too complicated to un-clarify our main idea that we want to present. Therefore, we focus on the single meaningful patch case in our work.*

To learn our structured data, we use two-layer convolutional neural network (CNN) (LeCun et al., 1998; Krizhevsky et al., 2012) with non-linear activation as the learner network.

**Definition 5.3.** (Two-layer Convolutional Network) For a given input data $\boldsymbol{X} \in \mathbb{R}^{Pd}$, our network learner $f_{\boldsymbol{W}} :$

$\mathbb{R}^{Pd} \to \mathbb{R}$ is defined as

$$f_{\boldsymbol{W}}(\boldsymbol{X}) = \sum_{r=1}^{m}\sum_{j=1}^{P} a_r^+ \sigma\left(\langle \boldsymbol{w}_r^+, \boldsymbol{X}[j]\rangle\right)$$
$$- \sum_{r=1}^{m}\sum_{j=1}^{P} a_r^- \sigma\left(\langle \boldsymbol{w}_r^-, \boldsymbol{X}[j]\rangle\right).$$

where $\boldsymbol{W} = [\{a_r^+\}_{r=1}^m, \{a_r^-\}_{r=1}^m, \{\boldsymbol{w}_r^+\}_{r=1}^m, \{\boldsymbol{w}_r^-\}_{r=1}^m]$ are learnable parameters, and $\sigma(\cdot)$ is the $\mathrm{ReLU}^q$ activation function defined as $\sigma(\cdot) = (\max(\cdot, 0))^q$ $(q \geq 2)$.

$\mathrm{ReLU}^q$ activation functions are useful to address the non-smoothness of ReLU function at zero, which are widely applied in literatures of deep learning theory (Kileel et al., 2019; Allen-Zhu & Li, 2020a;b; Chen et al., 2021; Jelassi & Li, 2022; Jelassi et al., 2022). To simplify our analysis, we fix the second layer weights as $a_r^+ = a_r^- = \frac{1}{2}$. We also assume that $q$ is odd and it holds that $w_r := w_r^+ = -w_r^-$ during optimization process. At initialization, we sample $\boldsymbol{w}_r$ i.i.d from $\mathcal{N}(0, \sigma_0^2\mathbf{I}_d)$. Our results can be extended to the case when $q$ is even and $w_r^+, -w_r^-$ are differently.

**The Role of Non-linearity.** Indeed, a series of recent theoretical works (Li et al., 2019; Chen et al., 2021; Javanmard & Soltanolkotabi, 2022) show that linear model can achieve robust generalization for adversarial training under certain settings, which but fails to explain the CGRO phenomenon observed in practice. To mitigate this gap, we improve the expressive power of model by using non-linear activation that can characterize the data structure and learning process more precisely.

In adversarial training, we aim to minimize the following adversarial training loss, which is a trade-off between natural risk and robust regularization defined as follow.

**Definition 5.4.** (Adversarial Training Loss) For a hyperparameter $\lambda > 0$ and the $\ell_p$-robust radius $\delta > 0$, the adversarial training loss of the network $f_{\boldsymbol{W}}$ w.r.t. the training dataset $\mathcal{S} = \{(\boldsymbol{X}_1, y_1), (\boldsymbol{X}_2, y_2), \ldots, (\boldsymbol{X}_N, y_N)\}$ is defined as

$$\widehat{\mathcal{L}}_{adv}(\boldsymbol{W}) := \frac{1}{N}\sum_{i=1}^{N}\underbrace{\mathcal{L}(\boldsymbol{W}; \boldsymbol{X}_i, y_i)}_{\text{natural risk}}$$
$$+ \lambda \cdot \underbrace{\max_{\widehat{\boldsymbol{X}}_i \in \mathbb{A}_p(\boldsymbol{X}_i, \delta)}\left[\mathcal{L}\left(\boldsymbol{W}; \widehat{\boldsymbol{X}}_i, y_i\right) - \mathcal{L}(\boldsymbol{W}; \boldsymbol{X}_i, y_i)\right]}_{\text{robust regularization}}.$$

where we use $\mathcal{L}(\boldsymbol{W}; \boldsymbol{X}, y)$ to denote the single-point loss with respect to $f_{\boldsymbol{W}}$ on $(\boldsymbol{X}, y)$ and $\mathbb{A}_p(\boldsymbol{X}, \delta)$ denotes the perturbed range of the data point $\boldsymbol{X}$ with $\ell_p$-radius $\delta$.

**Remark 5.5.** *Adversarial training loss in Definition 5.4 gives a general form of adversarial training methods (Goodfellow et al., 2014; Madry et al., 2017; Zhang et al., 2019) for different values of hyperparameter $\lambda$ and different types of loss function $\mathcal{L}(\boldsymbol{W}; \boldsymbol{X}, y)$ and perturbed range $\mathbb{A}_p(\cdot, \delta)$.*

Here, we use the logistic loss defined as $\mathcal{L}(\boldsymbol{W}; \boldsymbol{X}, y) := \log(1 + \exp\{-y f_W(\boldsymbol{X})\})$. And we apply the perturbed range defined as $\mathbb{A}_p(\boldsymbol{X}, \delta) := \mathbb{B}_p(\boldsymbol{X}, \delta) \cap \boldsymbol{X} + \boldsymbol{\Delta}(\boldsymbol{X})$, where $\boldsymbol{\Delta}(\boldsymbol{X}) \subset \mathbb{R}^{Pd}$ satisfies that

$$\boldsymbol{\Delta}(\boldsymbol{X})[j] = \begin{cases} \text{span}(\boldsymbol{w}^*) & \text{, if } j = \text{signal}(\boldsymbol{X}), \\ \boldsymbol{0} & \text{, otherwise.} \end{cases}$$

This perturbed range ensures that adversarial perturbations used to generate adversarial examples are always aligned with the meaningful signal vector $\boldsymbol{w}^*$ during adversarial training, which exactly simplifies the optimization analysis.

**Definition 5.6.** (Adversarial Training Algorithm) We run the standard gradient descent method to update the network parameters $\{\boldsymbol{W}^{(t)}\}_{t=0}^T$ for $T$ iterations w.r.t. the adversarial training loss that can be rewritten as $\widehat{\mathcal{L}}_{adv}^{(t)}(\boldsymbol{W}) = \frac{1}{N}\sum_{i=1}^N (1 - \lambda)\mathcal{L}(\boldsymbol{W}; \boldsymbol{X}_i, y_i) + \lambda \mathcal{L}\left(\boldsymbol{W}; \boldsymbol{X}_i^{adv,(t)}, y_i\right)$, where $\boldsymbol{X}_i^{adv,(t)}$ denotes the adversarial example generated by $i$-th training data $\boldsymbol{X}_i$ at $t$-th iteration. Concretely, the adversarial examples $\boldsymbol{X}_i^{adv,(t)}$ and network parameters $\boldsymbol{W}^{(t)}$ are updated alternatively as

$$\begin{cases} \boldsymbol{X}_i^{adv,(t)} = \underset{\widehat{\boldsymbol{X}}_i \in \mathbb{A}_p(\boldsymbol{X}_i, \delta)}{\text{argmax}} \mathcal{L}\left(\boldsymbol{W}^{(t)}; \widehat{\boldsymbol{X}}_i, y_i\right), i \in [N], \\ \boldsymbol{W}^{(t+1)} = \boldsymbol{W}^{(t)} - \eta \nabla_{\boldsymbol{W}} \widehat{\mathcal{L}}_{adv}^{(t)}\left(\boldsymbol{W}^{(t)}\right), \end{cases}$$

where $\eta > 0$ is the learning rate.

Next, we make the following assumptions about hyperparameters we introduced in our structured-data setting.

**Assumption 5.7.** (Choice of Hyperparameters) We assume that:

$$\alpha = \Theta(d^{c_\alpha}), \quad \sigma_p = \Theta(d^{-c_p}), \quad m = \Theta(N) = \text{poly}(d),$$

$$\sigma_0 = \frac{\text{polylog}(d)}{\sqrt{d}}, \quad \delta = \alpha\left(1 - \frac{1}{\sqrt{d}\,\text{polylog}(d)}\right),$$

$$\lambda \in \left[\frac{1}{\text{poly}(d)}, 1\right), \quad \eta = \left(0, \frac{1}{\text{poly}(d)}\right],$$

where $c_\alpha, c_p \in (0, 1)$ are constants satisfying $c_\alpha + c_p > \frac{1}{2}$.

**Discussion of Hyperparameter Choices.** Actually, the choices of these hyperparameters are not unique. We make concrete choices above for the sake of calculations in our proofs, but only the relationships between them are important. Namely, since the norm of signal patch is $\alpha$ and the norm of noise patch w.h.p. is $\Theta(\sigma_p\sqrt{d})$, our choices ensure that meaningful signal is stronger than noise. Without this assumption, in other word, if the signal-to-noise ratio is very low, there even exists no clean generalization, which has been theoretically shown under the similar patch-structured data setting (Cao et al., 2021; Chen et al., 2021; Frei et al., 2022). The width of network learner is $\tilde{O}(ND)$ to achieve mildly over-parameterization we mentioned in Theorem 4.4. The separation in Assumption 4.2 also holds due to $\delta < \alpha$.

## 5.2. Main Results

First, we introduce the concept called **feature learning** to characterize what the model learns from training dynamics.

**Definition 5.8.** (Feature Learning) Specifically, given a certain training data-point $(\boldsymbol{X}, y) \sim \mathcal{D}$ and the network learner $f_W$, we focus on the following two types of feature learning.

- **True Feature Learning.** We project the weight $\boldsymbol{W}$ on the meaningful signal vector to measure the correlation between the model and the true feature as

$$\mathcal{U} := \sum_{r=1}^m \langle \boldsymbol{w}_r, \boldsymbol{w}^* \rangle^q.$$

- **Spurious Feature Learning.** We project the weight $\boldsymbol{W}$ on the random noise to measure the correlation between the model and the spurious feature as

$$\mathcal{V} := \sum_{r=1}^m \sum_{j \in [P] \setminus \text{signal}(\boldsymbol{X})} \langle \boldsymbol{w}_r, y\boldsymbol{X}[j] \rangle^q.$$

Thus, it holds that the model correctly classify the clean data in two cases. One is that the model learns the true feature and ignores the spurious features, i.e. $\mathcal{U} = \Omega(1) \gg |\mathcal{V}|$. Another is that the model doesn't learn the true feature but memorizes the spurious features, i.e. $\mathcal{U} = o(1)$ and $\mathcal{V} = \Omega(1) \gg 0$. We can analyze the perturbed data similarly.

Now, we give our main result about feature learning process.

**Theorem 5.9.** *Under Assumption 5.7, we run the adversarial training algorithm in Definition 5.6 to update the weight of the network learner for $T = \Omega(\text{poly}(d))$ iterations. Then, with high probability, it holds that the network leanrer*

1. *partially learns the true feature, i.e. $\mathcal{U}^{(T)} = \Theta(\alpha^{-q})$;*

2. *exactly memorizes the spurious feature, i.e. for each $i \in [N], \mathcal{V}_i^{(T)} = \Theta(1)$,*

*where $\mathcal{U}^{(t)}$ and $\mathcal{V}_i^{(t)}$ is defined for $i$−th instance $(\boldsymbol{X}_i, y_i)$ and $t$−th iteration as the same in Definition 5.8. Consequently, the clean test error and robust training error are both smaller than $o(1)$, but the robust test error is at least $\frac{1}{2} - o(1)$, which means that $f_{\boldsymbol{W}^{(T)}}$ is a CGRO classifier.*

**Remark 5.10.** *Theorem 5.9 states that, during adversarial training, the neural network partially learns the true feature of objective classes and exactly memorizes the spurious features depending on specific training data, which causes that the network learner is able to correctly classify clean data by partial meaningful signal (clean generalization), but fails to classify the unseen perturbed data since it leverages only data-wise random noise to memorize training adversarial examples (robust overfitting). We also conduct numerical simulation experiments to confirm our results in Section 6.*

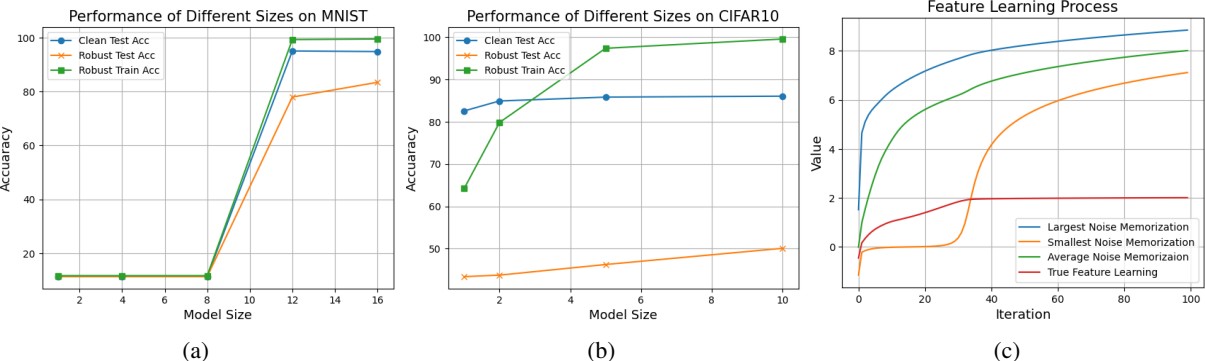

(a)  (b)  (c)

*Figure 2.* (a)(b): The effect of network capacity on the performance of the network. We trained the networks of varying capacity on MNIST (a) and CIFAR10 (b); (c): Feature learning process of the two-layer convolutional network on the structured data.

### 5.3. Analysis of Learning Process

Next, we provide a proof sketch of Theorem 5.9. To obtain a detailed analysis of learning process, we consider the following objects that can be viewed as weight-wise version of $\mathcal{U}^{(t)}$ and $\mathcal{V}_i^{(t)}$. We define $u^{(t)}$ and $v_{i,j}^{(t)}$ as

$$\begin{cases} u^{(t)} := \max_{r \in [m]} \left\langle \boldsymbol{w}_r^{(t)}, \boldsymbol{w}^* \right\rangle & (Signal\ Component), \\ v_{i,j}^{(t)} := \max_{r \in [m]} \left\langle \boldsymbol{w}_r^{(t)}, y_i \boldsymbol{X}_i[j] \right\rangle & (Noise\ Component), \end{cases}$$

for each $r \in [m]$, $i \in [N]$ and $j \in [P] \setminus \mathrm{signal}(\boldsymbol{X}_i)$.

We then propose a three-stage analysis technique to decouple the complicated feature learning process as follows.

**Phase I:** At the beginning, the signal component of lottery tickets winner $u^{(t)}$ increases quadratically (Lemma 5.11).

**Lemma 5.11.** *(Lower Bound of Signal Component Growth) For each $r \in [m]$ and any $t \geq 0$, the signal component grows as*

$$u^{(t+1)} \geq u^{(t)} + \Theta(\eta \alpha^q) \left( u^{(t)} \right)^{q-1} \psi \left( \alpha^q \mathcal{U}^{(t)} \right),$$

*where we use $\psi(\cdot)$ to denote the negative sigmoid function $\psi(z) = \frac{1}{1+e^z}$ as well as Lemma 5.12,5.13.*

Lemma 5.11 manifests that the signal component increases quadratically at initialization. Therefore, we know that, after $T_0 = \Theta \left( \eta^{-1} \alpha^{-q} \sigma_0^{-1} \right)$ iterations, the signal component $u^{(T_0)}$ attains the order $\tilde{\Omega}(\alpha^{-1})$, which implies the model learns partial true feature at this point.

**Phase II:** Once the signal component $u^{(t)}$ attains the order $\tilde{\Omega}(\alpha^{-1})$, the growth of signal component nearly stop updating since that the increment of signal component is now mostly dominated by the noise component (Lemma 5.12).

**Lemma 5.12.** *(Upper Bound of Signal Component Growth) For $T_0 = \Theta \left( \eta^{-1} \alpha^{-q} \sigma_0^{-1} \right)$ and any $t \in [T_0, T]$, the signal*

component is upper bounded as

$$u^{(t)} \leq \tilde{O} \left( \frac{\eta(\alpha-\delta)^q}{N} \right) \sum_{s=T_0}^{t-1} \sum_{i=1}^{N} \psi \left( (\alpha-\delta)^q \mathcal{U}^{(s)} + \mathcal{V}_i^{(s)} \right) + \tilde{O}(\alpha^{-1}).$$

Lemma 5.12 shows that, after partial true feature learning, the increment of signal component is mostly dominated by the noise component $\mathcal{V}_i^{(t)}$, which implies that the growth of signal component will converge when $\mathcal{V}_i^{(t)} = \Omega(1)$.

**Phase III:** After that, by the quadratic increment of noise component (Lemma 5.13), the total noise $\mathcal{V}_i^{(t)}$ eventually attains the order $\Omega(1)$, which implies the model memorizes the spurious feature (data-wise noise) in final.

**Lemma 5.13.** *(Lower Bound of Noise Component Growth) For each $i \in [N]$, $r \in [m]$ and $j \in [P] \setminus \mathrm{signal}(\boldsymbol{X}_i)$ and any $t \geq 1$, the noise component grows as*

$$v_{i,j}^{(t)} \geq v_{i,j}^{(0)} + \Theta \left( \frac{\eta \sigma_p^2 d}{N} \right) \sum_{s=0}^{t-1} \psi(\mathcal{V}_i^{(s)}) \left( v_{i,j}^{(s)} \right)^{q-1} - \tilde{O}(P \sigma_p^2 \alpha^{-1} \sqrt{d}).$$

The practical implication of Lemma 5.13 is two-fold. First, by the quadratic increment of noise component, we derive that, after $T_1 = \Theta \left( N \eta^{-1} \sigma_0^{-1} \sigma_p^{-3} d^{-\frac{3}{2}} \right)$ iterations, the total noise memorization $\mathcal{V}_i^{(T)}$ attains the order $\Omega(1)$, which suggests that the model is able to robustly classify adversarial examples by memorizing the data-wise noise. Second, combined with Lemma 5.12, the signal component $u^{(t)}$ will maintain the order $\Theta(\alpha^{-1})$, which immediately implies the main conclusion of Theorem 5.9. And the full detailed proof of Theorem 5.9 can be see in Appendix E.

*Table 1.* Performance of Models with Different Sizes

| Dataset | MNIST | | | | | CIFAR10 | | | |
|---|---|---|---|---|---|---|---|---|---|
| **Model Size Factor** | 1 | 2 | 8 | 12 | 16 | 1 | 2 | 5 | 10 |
| **Clean Test Acc** | 11.35 | 11.35 | 11.35 | 95.06 | 94.85 | 82.56 | 84.92 | 85.83 | 86.05 |
| **Robust Test Acc** | 11.35 | 11.35 | 11.35 | 77.96 | 83.43 | 43.39 | 43.74 | 46.25 | 50.08 |
| **Robust Train Acc** | 11.70 | 11.70 | 11.70 | 99.3 | 99.5 | 64.19 | 79.82 | 97.37 | 99.57 |

## 6. Experiments

In this section, we empirically verify our theoretical results in Section 4 and Section 5 by experiments in real-world image-recognition datasets and synthetic structured data.

### 6.1. Effect of Different Model Sizes

**Experiment Settings.** For the MNIST dataset, we consider a simple convolutional network, LeNet-5 (LeCun et al., 1998), and study how its performance changes as we increases the size of network (i.e. we expand the number of convolutional filters and the size of the fully connected layer to the size number multiple of the initial, where the size numbers are $1, 4, 8, 12, 16$). The original convolutional network has a convolutional layer with $1$ filters, followed by another convolutional layer with $2$ filters, and a fully connected hidden layer with $32$ units. Convolutional layers are followed by $2 \times 2$ max-pooling layers. For the CIFAR10 dataset, we apply WideResNet-34 (Zagoruyko & Komodakis, 2016) with different widen factors $1, 2, 5, 10$. We use the standard projected gradient descent (PGD) (Madry et al., 2017) to train the network by adversarial training. We choose the classical $\ell_\infty$-perturbation with radius $0.3$ for MNIST and $8/255$ for CIFAR10. All models are trained for 100 epoches on MNIST and 200 epoches on CIFAR10.

**Experiment Results.** We report our results about the performance (including the clean test accuracy, robust test accuracy and robust training accuracy) of models with different sizes in Table 1 and Figure 2 (a)(b). It shows that when the model size becomes larger, first the robust training loss decreases but the robust generalization gap remains large, and then when the model gets even larger, the robust generalization gap gradually decreases, and we also find that, in the small size case (see LeNet with the size number $1, 4, 8$), adversarial training converges to a trivia solution that always predicts a fixed class, while it can learn an accurate clean classifier through the standard training, which corresponds to the theoretical results in Theorem 4.4 and Theorem 4.7.

### 6.2. Synthetic Structured Data

**Experiment Settings.** Here we generate synthetic data exactly following Definition 5.1 and apply the two-layer convolutional network in Definition 5.3. We train the network

by the adversarial training algorithm we mentioned in Definition 5.6. We choose the hyperparameters that we need as: $d = 100, P = 2, \alpha = 10, \sigma_p = 1, \sigma_0 = 0.01, q = 3, N = 20, m = 10, p = 2, \delta = 10, \lambda = 0.9, \eta = 0.1, T = 100$, which is a feasible one under Assumption 5.7. We characterize the true feature learning and noise memorization via calculating $\mathcal{U}^{(t)}$ and the smallest/largest/average of $\{\mathcal{V}_i^{(t)}\}_{i \in [N]}$. We calculate the robust train and test accuracy of the model by using the standard PGD attack.

*Table 2.* Performance on Synthetic Data

| | Train | test |
|---|---|---|
| **Clean Acc** | 100.0 | 98.5 |
| **Robust Acc** | 100.0 | 17.5 |

**Experiment Results.** We plot the dynamics of true feature learning and noise memorization in Figure 2 (c). It is clear that a three-stage phase transitions happen during adversarial training , which is consistent with our theoretical analysis of learning process (Lemma 5.11, Lemma 5.12 and Lemma 5.13), and finally the model partially learns true feature but exactly memorizes all training data (Theorem 5.9). The performance of the model is presented in Table 2, which shows that the network converges to a CGRO classifier eventually.

## 7. Conclusion

In this paper, we study the CGRO phenomenon in adversarial training and present theoretical explanations: from the perspective of representation complexity, we prove that the CGRO classifier is efficient to achieve by leveraging robust memorization regarding the training data, while robust generalization requires excessively large model capacity in worst case, which may lead adversarial training to the CGRO regime; from the perspective of training dynamics, we propose a three-stage analysis technique to analyze the feature learning process of adversarial training under our structured-data framework, and it shows that two-layer neural network trained by adversarial training provably learns the partial true feature but memorizes the random noise from training data, which thereby causes the CGRO regime. On the empirical side, we confirm our theoretical findings above by real-world vision datasets and synthetic data simulation.

## Acknowledgements

Binghui Li is partially supported by the Elite Ph.D. Program in Applied Mathematics for PhD Candidates in Peking University. We thank anonymous reviewers for their valuable suggestions.

## Impact Statement

This paper presents work whose goal is to advance the field of Machine Learning. There are many potential societal consequences of our work, none which we feel must be specifically highlighted here.

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

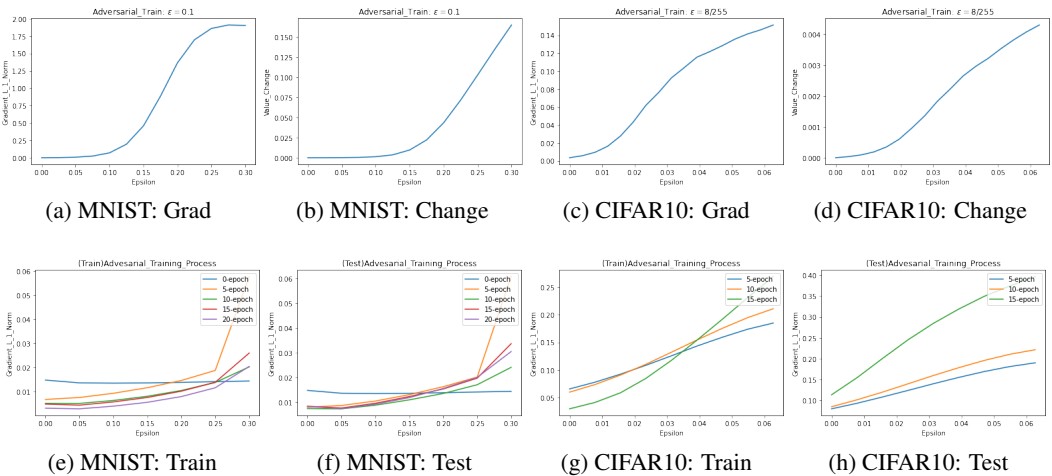

Figure 3. Experiment Results ($\ell_\infty$ Perturbation Radius $\epsilon_0 = 0.1$ on MNIST, $= 8/255$ on CIFAR10).

## A. Additional Experiments Regarding Robust Memorization

In this section, we demonstrate that adversarial training converges to similarities of the construction $f_S$ of Lemma 4.5 on real image datasets, which results in CGRO. In fact, we need to verify models trained by adversarial training tend to memorize data by approximating local robust indicators on training data.

Concretely, for given loss $\mathcal{L}(\cdot, \cdot)$, instance $(\boldsymbol{X}, y)$ and model $f$, we use two measurements, maximum gradient norm within the neighborhood of training data,

$$\max_{\|\boldsymbol{\xi}\|_\infty \leq \delta} \|\nabla_{\boldsymbol{X}} \mathcal{L}(f(\boldsymbol{X} + \boldsymbol{\xi}), y)\|_1,$$

and maximum loss function value change

$$\max_{\|\boldsymbol{\xi}\|_\infty \leq \delta} [\mathcal{L}(f(\boldsymbol{X} + \boldsymbol{\xi}), y) - \mathcal{L}(f(\boldsymbol{X}), y)]$$

.

The former measures the $\delta-$local flatness on $(\boldsymbol{X}, y)$, and the latter measures $\delta-$local adversarial robustness on $(\boldsymbol{X}, y)$, which both describe the key information of loss landscape over input.

**Experiment Settings.** In numerical experiments, we mainly focus on two common real-image datasets, MNIST and CIFAR10. During adversarial training, we use cyclical learning rates and mixed precision technique (Wong et al., 2020). For the MNIST dataset, we use a LeNet architecture and train total 20 epochs. For the CIFAR10 dataset, we use a Resnet architecture and train total 15 epochs.

**Numerical Results.** First, we apply the adversarial training method to train models by a fixed perturbation radius $\epsilon_0$, and then we compute empirical average of maximum gradient norm and maximum loss change on training data within different perturbation radius $\epsilon$. We can see numerical results in Figure 3 (a∼d), and it shows that loss landscape has flatness within the training radius, but is very sharp outside, which practically demonstrates our conjecture on real image datasets.

**Learning Process.** We also focus on the dynamics of loss landscape over input during the adversarial learning process. Thus, we compute empirical average of maximum gradient norm within different perturbation radius $\epsilon$ and in different training epochs. The numerical results are plotted in Figure 3 (e∼h). Both on MNIST and CIFAR10, with epochs increasing, it is observed that the training curve descents within training perturbation radius, which implies models learn the local robust indicators to robustly memorize training data. However, the test curve of CIFAR10 ascents within training radius instead, which is consistent with our theoretical analysis in Section 5.

**Robust Generalization Bound.** Moreover, we prove a robust generalization bound based on *global flatness* of loss landscape (see in Appendix B). We show that, while adversarial training achieves local flatness by robust memorization, the model lacks global flatness, which causes robust overfitting.

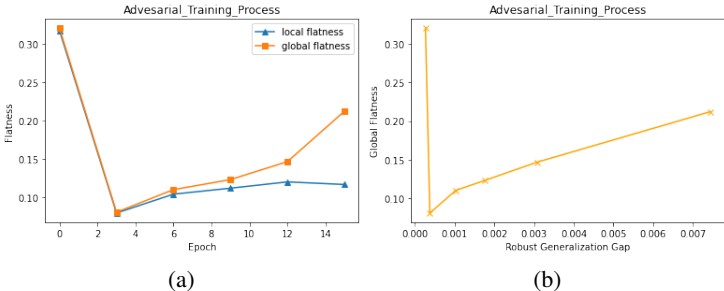

(a)                 (b)

*Figure 4.* **Left:** Local and Global Flatness During Adversarial Training on CIFAR10; **Right:** The Relation Between Robust Generalization Gap and Global Flatness on CIFAR10.

## B. Robust Generalization Bound Based on Global Flatness

In this section, we prove a novel robust generalization bound that mainly depends on global flatness of loss landscape. We consider $\ell_p-$adversarial robustness with perturbation radius $\delta$ and we use $\mathcal{L}_{\text{clean}}$, $\mathcal{L}_{\text{adv}}(f)$ and $\widehat{\mathcal{L}}_{\text{adv}}(f)$ to denote the clean test risk, the adversarial test risk and the adversarial empirical risk w.r.t. the model $f$, respectively. We also assume $\frac{1}{p} + \frac{1}{q} = 1$ for the next results.

**Theorem B.1.** *(Robust Generalization Bound) Let $\mathcal{D}$ be the underlying distribution with a smooth density function, and $N-$sample training dataset $\mathcal{S} = \{(\boldsymbol{X}_1, y_1), (\boldsymbol{X}_2, y_2), \ldots, (\boldsymbol{X}_N, y_N)\}$ is i.i.d. drawn from $\mathcal{D}$. Then, with high probability, it holds that,*

$$\mathcal{L}_{adv}(f) \leq \widehat{\mathcal{L}}_{adv}(f) + N^{-\frac{1}{D+2}} O \left( \underbrace{\mathbb{E}_{(\boldsymbol{X},y) \sim \mathcal{D}} \left[ \max_{\|\boldsymbol{\xi}\|_p \leq \delta} \|\nabla_{\boldsymbol{X}} \mathcal{L}(f(\boldsymbol{X} + \boldsymbol{\xi}), y)\|_q \right]}_{\text{global flatness}} \right).$$

This generalization bound shows that robust generalization gap can be dominated by global flatness of loss landscape. And we also have the lower bound of robust generalization gap stated as follow.

**Proposition B.2.** *Let $\mathcal{D}$ be the underlying distribution with a smooth density function, then we have*

$$\mathcal{L}_{adv}(f) - \mathcal{L}_{clean}(f) = \Omega \left( \delta \mathbb{E}_{(\boldsymbol{X},y) \sim \mathcal{D}} \left[ \|\nabla_{\boldsymbol{X}} \mathcal{L}(f(\boldsymbol{X}), y)\|_q \right] \right).$$

Theorem B.1 and Proposition B.2 manifest that robust generalization gap is very related to global flatness. However, although adversarial training achieves good local flatness by robust memorization on training data, the model lacks global flatness, which leads to robust overfitting.

This point is also verified by numerical experiment on CIFAR10 (see results in Figure 4). First, global flatness grows much faster than local flatness in practice. Second, with global flatness increasing during training process, it causes an increase of robust generalization gap.

# C. Preliminary Lemmas

First, we present a technique called *Tensor Power Method* proposed by Allen-Zhu & Li (2020a;b).

**Lemma C.1.** *Let* $\left\{z^{(t)}\right\}_{t=0}^{T}$ *be a positive sequence defined by the following recursions*

$$\begin{cases} z^{(t+1)} \geq z^{(t)} + m\left(z^{(t)}\right)^2 \\ z^{(t+1)} \leq z^{(t)} + M\left(z^{(t)}\right)^2 \end{cases},$$

*where* $z^{(0)} > 0$ *is the initialization and* $m, M > 0$. *Let* $v > 0$ *such that* $z^{(0)} \leq v$. *Then, the time* $t_0$ *such that* $z_t \geq v$ *for all* $t \geq t_0$ *is:*

$$t_0 = \frac{3}{mz^{(0)}} + \frac{8M}{m}\left\lceil \frac{\log\left(v/z_0\right)}{\log(2)} \right\rceil.$$

**Lemma C.2.** *Let* $\left\{z^{(t)}\right\}_{t=0}^{T}$ *be a positive sequence defined by the following recursion*

$$\begin{cases} z^{(t)} \geq z^{(0)} + A\sum_{s=0}^{t-1}\left(z^{(s)}\right)^2 - C \\ z^{(t)} \leq z^{(0)} + A\sum_{s=0}^{t-1}\left(z^{(s)}\right)^2 + C \end{cases}$$

*where* $A, C > 0$ *and* $z^{(0)} > 0$ *is the initialization. Assume that* $C \leq z^{(0)}/2$. *Let* $v > 0$ *such that* $z^{(0)} \leq v$. *Then, the time* $t$ *such that* $z^{(t)} \geq v$ *is upper bounded as:*

$$t_0 = 8\left\lceil \frac{\log\left(v/z_0\right)}{\log(2)} \right\rceil + \frac{21}{\left(z^{(0)}\right)A}.$$

**Lemma C.3.** *Let* $\mathcal{T} \geq 0$. *Let* $(z_t)_{t>\mathcal{T}}$ *be a non-negative sequence that satisfies the recursion:* $z^{(t+1)} \leq z^{(t)} - A\left(z^{(t)}\right)^2$, *for* $A > 0$. *Then, it is bounded at a time* $t > \mathcal{T}$ *as*

$$z^{(t)} \leq \frac{1}{A(t-\mathcal{T})}.$$

Then, we provide a probability inequality proved by Jelassi & Li (2022).

**Lemma C.4.** *Let* $\{v_r\}_{r=1}^{m}$ *be vectors in* $\mathbb{R}^d$ *such that there exist a unit norm vector* $x$ *that satisfies* $\left|\sum_{r=1}^{m}\langle v_r, x\rangle^3\right| \geq 1$. *Then, for* $\xi_1, \ldots, \xi_k \sim \mathcal{N}\left(0, \sigma^2 \mathbf{I}_d\right)$ *i.i.d., we have:*

$$\mathbb{P}\left[\left|\sum_{j=1}^{P}\sum_{r=1}^{m}\langle v_r, \xi_j\rangle^3\right| \geq \tilde{\Omega}\left(\sigma^3\right)\right] \geq 1 - \frac{O(d)}{2^{1/d}}.$$

Next, we introduce some concepts about learning theory.

**Definition C.5** (growth function). Let $\mathcal{F}$ be a class of functions from $\mathcal{X} \subset \mathbb{R}^d$ to $\{-1, +1\}$. For any integer $m \geq 0$, we define the growth function of $\mathcal{F}$ to be

$$\Pi_{\mathcal{F}}(m) = \max_{x_i \in \mathcal{X}, 1 \leq i \leq m}\left|\{(f(x_1), f(x_2), \cdots, f(x_m)) : f \in \mathcal{F}\}\right|.$$

In particular, if $\left|\{(f(x_1), f(x_2), \cdots, f(x_m)) : f \in \mathcal{F}\}\right| = 2^m$, then $(x_1, x_2, \cdots, x_m)$ is said to be *shattered* by $\mathcal{F}$.

**Definition C.6** (Vapnik-Chervonenkis dimension). Let $\mathcal{F}$ be a class of functions from $\mathcal{X} \subset \mathbb{R}^D$ to $\{-1, +1\}$. The VC-dimension of $\mathcal{F}$, denoted by VC-dim($\mathcal{F}$), is defined as the largest integer $m \geq 0$ such that $\Pi_{\mathcal{F}}(m) = 2^m$. For real-value function class $\mathcal{H}$, we define VC-dim($\mathcal{H}$) := VC-dim(sgn($\mathcal{H}$)).

The following result gives a nearly-tight upper bound on the VC-dimension of neural networks.

**Lemma C.7.** *(Bartlett et al., 2019) Consider a ReLU network with* $L$ *layers and* $W$ *total parameters. Let* $F$ *be the set of (real-valued) functions computed by this network. Then we have VC-dim($F$) = $O(WL\log(W))$.*

The growth function is connected to the VC-dimension via the following lemma; see e.g. Anthony et al. (1999).

**Lemma C.8.** *Suppose that VC-dim($\mathcal{F}$) = $k$, then* $\Pi_m(\mathcal{F}) \leq \sum_{i=0}^{k}\binom{m}{i}$. *In particular, we have* $\Pi_m(\mathcal{F}) \leq (em/k)^k$ *for all* $m > k + 1$.

# D. Feature Learning

In this section, we provide a full introduction to feature learning, which is widely applied in theoretical works (Allen-Zhu & Li, 2020a;b; 2022; Shen et al., 2022; Jelassi & Li, 2022; Jelassi et al., 2022; Chen et al., 2022) explore what and how neural networks learn in different tasks. In this work, we first leverage feature learning theory to explain CGRO phenomenon in adversarial training. Specifically, for an arbitrary clean training data-point $(\boldsymbol{X}, y) \sim \mathcal{D}$ and a given model $f_{\boldsymbol{W}}$, we focus on

- **True Feature Learning.** We project the weight $\boldsymbol{W}$ on the meaningful signal vector to measure the correlation between the model and the true feature as

$$\mathcal{U} := \sum_{r=1}^{m} \langle \boldsymbol{w}_r, \boldsymbol{w}^* \rangle^q .$$

- **Spurious Feature Learning.** We project the weight $\boldsymbol{W}$ on the random noise to measure the correlation between the model and the spurious feature as

$$\mathcal{V} := y \sum_{r=1}^{m} \sum_{j \in [P] \backslash \mathrm{signal}(\boldsymbol{X})} \langle \boldsymbol{w}_r, \boldsymbol{X}[j] \rangle^q .$$

We then calculate the model's classification correctness on certain clean data point as

$$
\begin{aligned}
y f_{\boldsymbol{W}}(\boldsymbol{X}) &= y \sum_{r=1}^{m} \langle \boldsymbol{w}_r, \boldsymbol{X}[\mathrm{signal}(\boldsymbol{X})] \rangle^q + y \sum_{r=1}^{m} \sum_{j \in [P] \backslash \mathrm{signal}(\boldsymbol{X})} \langle \boldsymbol{w}_r, \boldsymbol{X}[j] \rangle^q \\
&= \underbrace{y \sum_{r=1}^{m} \langle \boldsymbol{w}_r, \alpha y \boldsymbol{w}^* \rangle^q}_{\alpha^q \mathcal{U}} + \underbrace{y \sum_{r=1}^{m} \sum_{j \in [P] \backslash \mathrm{signal}(\boldsymbol{X})} \langle \boldsymbol{w}_r, \boldsymbol{X}[j] \rangle^q}_{\mathcal{V}} .
\end{aligned}
$$

Thus, the model correctly classify the data if and only if $\alpha^q \mathcal{U} + \mathcal{V} \geq 0$, which holds in at least two cases. Indeed, one is that the model learns the true feature and ignores the spurious features, where $\mathcal{U} = \Omega(1) \gg |\mathcal{V}|$. Another is that the model doesn't learn the true feature but memorizes the spurious features, where $|\mathcal{U}| = o(1)$ and $|\mathcal{V}| = \Omega(1) \gg 0$.

Therefore, this analysis tells us that the model will generalize well for unseen data if the model learns true feature learning. But the model will overfit training data if the model only memorizes spurious features since the data-specific random noises are independent for distinct instances, which means that, with high probability, it holds that $\mathcal{V} = o(1)$ for unseen data $(\boldsymbol{X}, y)$.

We also calculate the model's classification correctness on perturbed data point, where we use attack proposed in definition 5.6 to generate adversarial example as

$$
\boldsymbol{X}^{\mathrm{adv}}[j] = \begin{cases} \alpha(1 - \gamma) y \boldsymbol{w}^*, & j = \mathrm{signal}(\boldsymbol{X}) \\ \boldsymbol{X}[j], & j \in [p] \backslash \mathrm{signal}(\boldsymbol{X}) \end{cases}
$$

We then derive the correctness as

$$
\begin{aligned}
y f_{\boldsymbol{W}}(\boldsymbol{X}^{\mathrm{adv}}) &= y \sum_{r=1}^{m} \langle \boldsymbol{w}_r, \boldsymbol{X}^{\mathrm{adv}}[\mathrm{signal}(\boldsymbol{X})] \rangle^q + y \sum_{r=1}^{m} \sum_{j \in [P] \backslash \mathrm{signal}(\boldsymbol{X})} \langle \boldsymbol{w}_r, \boldsymbol{X}^{\mathrm{adv}}[j] \rangle^q \\
&= \underbrace{y \sum_{r=1}^{m} \langle \boldsymbol{w}_r, \alpha(1 - \gamma) y \boldsymbol{w}^* \rangle^q}_{\alpha^q (1-\gamma)^q \mathcal{U}} + \underbrace{y \sum_{r=1}^{m} \sum_{j \in [P] \backslash \mathrm{signal}(\boldsymbol{X})} \langle \boldsymbol{w}_r, \boldsymbol{X}[j] \rangle^q}_{\mathcal{V}} .
\end{aligned}
$$

Thus, the model correctly classify the perturbed data if and only if $\alpha^q (1 - \gamma)^q \mathcal{U} + \mathcal{V} \geq 0$, which implies that We can analyze the perturbed data similarly.

# E. Proof for Section 5

In this section, we present the full proof for Section 5. To simplify our proof, we only focus on the case when $q = 3$, and it can be easily extended to the general case when $q \geq 2$. And we need the re-defined notation as the following.

For $r \in [m]$, $i \in [N]$ and $j \in [P] \setminus \text{signal}(\boldsymbol{X}_i)$, we define $u_r^{(t)}$ and $v_{i,j,r}^{(t)}$ as

**Signal Component.** $u_r^{(t)} := \left\langle \boldsymbol{w}_r^{(t)}, \boldsymbol{w}^* \right\rangle$, thus $\mathcal{U}^{(t)} = \sum_{r \in [m]} \left( u_r^{(t)} \right)^3$.

**Noise Component.** $v_{i,j,r}^{(t)} := y_i \left\langle \boldsymbol{w}_r^{(t)}, \boldsymbol{X}_i[j] \right\rangle$, thus $\mathcal{V}_i^{(t)} = \sum_{r \in [m]} \sum_{j \in [P] \setminus \text{signal}(\boldsymbol{X}_i)} \left( v_{i,j,r}^{(t)} \right)^3$.

First, we give detailed proofs of Lemma 5.11, Lemma 5.12 and Lemma 5.13. Then, we prove Theorem 5.9 base on the above lemmas.

We prove our main results using an induction. More specifically, we make the following assumptions for each iteration $t < T$.

**Hypothesis E.1.** *Throughout the learning process using the adversarial training update for $t < T$, we maintain that:*

- *(Uniform Bound for Signal Component) For each $r \in [m]$, we assume $u_r^{(t)} \leq \tilde{O}(\alpha^{-1})$.*

- *(Uniform Bound for Noise Component) For each $r \in [m]$, $i \in [N]$ and $j \in [P] \setminus \text{signal}(\boldsymbol{X}_i)$, we assume $|v_{i,j,r}^{(t)}| \leq \tilde{O}(1)$.*

In what follows, we assume these induction hypotheses for $t < T$ to prove our main results. We then prove these hypotheses for iteration $t = T$ in Lemma E.11.

Now, we first give proof details about Lemma 5.11.

**Theorem E.2.** *(Restatement of Lemma 5.11) For each $r \in [m]$ and any $t \geq 0$, the signal component grows as*

$$u_r^{(t+1)} \geq u_r^{(t)} + \Theta(\eta \alpha^3) \left( u_r^{(t)} \right)^2 \psi \left( \alpha^3 \sum_{k \in [m]} \left( u_k^{(t)} \right)^3 \right),$$

*where we use $\psi(\cdot)$ to denote the negative sigmoid function $\psi(z) = \frac{1}{1+e^z}$ as well as Lemma 5.12,5.13.*

*Proof.* First, we calculate the gradient of adversarial loss with respect to $\boldsymbol{w}_r (r \in [m])$ as

$$\nabla_{\boldsymbol{w}_r} \widehat{\mathcal{L}}_{\text{adv}}(\boldsymbol{W}^{(t)}) = -\frac{3}{N} \sum_{i=1}^N \sum_{j=1}^P \left( \frac{(1-\lambda)y_i \left\langle \boldsymbol{w}_r^{(t)}, \boldsymbol{X}_i[j] \right\rangle^2}{1 + \exp\left(y_i f_{\boldsymbol{W}^{(t)}}\left(\boldsymbol{X}_i\right)\right)} \boldsymbol{X}_i[j] + \frac{\lambda y_i \left\langle \boldsymbol{w}_r^{(t)}, \boldsymbol{X}_i^{\text{adv}}[j] \right\rangle^2}{1 + \exp\left(y_i f_{\boldsymbol{W}^{(t)}}\left(\boldsymbol{X}_i^{\text{adv}}\right)\right)} \boldsymbol{X}_i^{\text{adv}}[j] \right)$$

$$= -\frac{3}{N} \left( \left( u_r^{(t)} \right)^2 \left( \sum_{i=1}^N (1-\lambda)\alpha^3 \psi(y_i f_{\boldsymbol{W}^{(t)}}(\boldsymbol{X}_i)) + \lambda \alpha^3 (1-\gamma)^3 \psi(y_i f_{\boldsymbol{W}^{(t)}}(\boldsymbol{X}_i^{\text{adv}})) \right) \boldsymbol{w}^* \right.$$

$$\left. + \sum_{i=1}^N \sum_{j \neq \text{signal}(\boldsymbol{X}_i)} \left( v_{i,j,r}^{(t)} \right)^2 \left( (1-\lambda)\psi(y_i f_{\boldsymbol{W}^{(t)}}(\boldsymbol{X}_i)) + \lambda \psi(y_i f_{\boldsymbol{W}^{(t)}}(\boldsymbol{X}_i^{\text{adv}})) \right) \boldsymbol{X}_i[j] \right).$$

Then, we project the gradient descent algorithm equation $\boldsymbol{W}^{(t+1)} = \boldsymbol{W}^{(t)} - \eta \nabla_{\boldsymbol{W}} \widehat{\mathcal{L}}_{\text{adv}}\left(\boldsymbol{W}^{(t)}\right)$ on the signal vector $\boldsymbol{w}^*$.

We derive the following result due to $\boldsymbol{X}_i[j] \perp \boldsymbol{w}^*$ for $j \in [P] \setminus \mathrm{signal}(\boldsymbol{X}_i)$.

$$u_r^{(t+1)} = u_r^{(t)} + \frac{3\eta}{N} \left(u_r^{(t)}\right)^2 \sum_{i=1}^{N} \left((1-\lambda)\alpha^3 \psi(y_i f_{\boldsymbol{W}^{(t)}}(\boldsymbol{X}_i)) + \lambda\alpha^3(1-\gamma)^3 \psi(y_i f_{\boldsymbol{W}^{(t)}}(\boldsymbol{X}_i^{\mathrm{adv}})))\right)$$

$$\geq u_r^{(t)} + \frac{3\eta\alpha^3(1-\lambda)}{N} \left(u_r^{(t)}\right)^2 \sum_{i=1}^{N} \psi(y_i f_{\boldsymbol{W}^{(t)}}(\boldsymbol{X}_i))$$

$$\geq u_r^{(t)} + \Theta(\eta\alpha^3) \left(u_r^{(t)}\right)^2 \psi\left(\alpha^3 \sum_{k \in [m]} \left(u_k^{(t)}\right)^3\right),$$

where we derive last inequality by using $\psi(y_i f_{\boldsymbol{W}^{(t)}}(\boldsymbol{X}_i)) = \Theta(1)\psi\left(\alpha^3 \sum_{k \in [m]} \left(u_k^{(t)}\right)^3\right)$, which is obtained due to Hypothesis E.1.

$\square$

Consequently, we have the following result that shows the order of maximum signal component.

**Lemma E.3.** *During adversarial training, with high probability, it holds that, after $T_0 = \tilde{\Theta}\left(\frac{1}{\eta\alpha^3\sigma_0}\right)$ iterations, for all $t \in [T_0, T]$, we have $\max_{r \in [m]} u_r^{(t)} \geq \tilde{\Omega}(\alpha^{-1})$.*

*Proof.* From the proof of Theorem E.2, we know that

$$u_r^{(t+1)} - u_r^{(t)} = \frac{3\eta}{N} \left(u_r^{(t)}\right)^2 \sum_{i=1}^{N} \left((1-\lambda)\alpha^3 \psi(y_i f_{\boldsymbol{W}^{(t)}}(\boldsymbol{X}_i)) + \lambda\alpha^3(1-\gamma)^3 \psi(y_i f_{\boldsymbol{W}^{(t)}}(\boldsymbol{X}_i^{\mathrm{adv}})))\right).$$

By applying Hypothesis E.1, we can simplify the above equation to the following inequalities.

$$\begin{cases} u_r^{(t+1)} \leq u_r^{(t)} + A \left(u_r^{(t)}\right)^2 \\ u_r^{(t+1)} \geq u_r^{(t)} + B \left(u_r^{(t)}\right)^2 \end{cases}$$

where $A$ and $B$ are respectively defined as:

$$A := \tilde{\Theta}(\eta) \left((1-\lambda)\alpha^3 + \lambda\alpha^3(1-\gamma)^3\right)$$
$$B := \tilde{\Theta}(\eta)(1-\lambda)\alpha^3.$$

At initialization, since we choose the weights $\boldsymbol{w}_r^{(0)} \sim \mathcal{N}\left(0, \sigma_0^2 \mathbf{I}_d\right)$, we know the initial signal components $u_r^{(0)}$ are i.i.d. zero-mean Gaussian random variables, which implies that the probability that at least one of the $u_r^{(0)}$ is non-negative is $1 - \left(\frac{1}{2}\right)^m = 1 - o(1)$.

Thus, with high probability, there exists an initial signal component $u_{r'}^{(0)} \geq 0$. By using Tensor Power Method (Lemma C.1) and setting $v = \tilde{\Theta}(\alpha^{-1})$, we have the threshold iteration $T_0$ as

$$T_0 = \frac{\tilde{\Theta}(1)}{\eta\alpha^3\sigma_0} + \frac{\tilde{\Theta}(1)\left((1-\lambda)\alpha^3 + \lambda\beta^3\right)}{(1-\lambda)\alpha^3} \left\lceil \frac{-\log\left(\tilde{\Theta}(\sigma_0\alpha)\right)}{\log(2)} \right\rceil.$$

$\square$

Next, we prove Lemma 5.12 to give an upper bound of signal components' growth.

**Theorem E.4.** *(Restatement of Lemma 5.12) For $T_0 = \Theta\left(\frac{1}{\eta\alpha^3\sigma_0}\right)$ and any $t \in [T_0, T]$, the signal component is upper bounded as*

$$\max_{r\in[m]} u_r^{(t)} \leq \tilde{O}(\alpha^{-1}) + \tilde{O}\left(\frac{\eta\alpha^3(1-\gamma)^3}{N}\right) \sum_{s=T_0}^{t-1}\sum_{i=1}^{N} \psi\left(\alpha^3(1-\gamma)^3 \sum_{k\in[m]}\left(u_k^{(s)}\right)^3 + \mathcal{V}_i^{(s)}\right).$$

*Proof.* First, we analyze the upper bound of derivative generated by clean data. By following the proof of Theorem E.2, we know that, for each $r \in [m]$,

$$\max_{r\in[m]} u_r^{(t+1)} \geq \max_{r\in[m]} u_r^{(t)} + \frac{3\eta\alpha^3(1-\lambda)}{N}\left(\max_{r\in[m]} u_r^{(t)}\right)^2 \sum_{i=1}^{N}\psi(y_i f_{\boldsymbol{W}^{(t)}}(\boldsymbol{X}_i))$$

$$\geq \max_{r\in[m]} u_r^{(t)} + \tilde{\Omega}(\eta\alpha)\frac{1-\lambda}{N}\sum_{i=1}^{N}\psi(y_i f_{\boldsymbol{W}^{(t)}}(\boldsymbol{X}_i)),$$

where we obtain the first inequality by the definition of $\max_{r\in[m]} u_r^{(t)}, \max_{r\in[m]} u_r^{(t+1)}$, and we use $\max_{r\in[m]} u_r^{(t)} \geq \tilde{\Omega}(\alpha^{-1})$ derived by Lemma E.3 in the last inequality. Thus, we then have

$$\frac{1-\lambda}{N}\sum_{i=1}^{N}\psi(y_i f_{\boldsymbol{W}^{(t)}}(\boldsymbol{X}_i)) \leq \tilde{O}(\eta^{-1}\alpha^{-1})\left(\max_{r\in[m]} u_r^{(t+1)} - \max_{r\in[m]} u_r^{(t)}\right).$$

Now, we focus on $\max_{r\in[m]} u_r^{(t+1)} - \max_{r\in[m]} u_r^{(t)}$. By the non-decreasing property of $u_r^{(t)}$, we have

$$\max_{r\in[m]} u_r^{(t+1)} - \max_{r\in[m]} u_r^{(t)} \leq \sum_{r\in[m]}\left(u_r^{(t+1)} - u_r^{(t)}\right)$$

$$\leq (1-\lambda)\Theta(\eta\alpha)\psi\left(\alpha^3\sum_{r\in[m]}\left(u_r^{(t)}\right)^3\right)\sum_{r\in[m]}\left(\alpha u_r^{(t)}\right)^2 + \lambda\Theta\left(\frac{\eta\alpha^3(1-\gamma)^3}{N}\right)\sum_{i=1}^{N}\psi(y_i f_{\boldsymbol{W}^{(t)}}(\boldsymbol{X}_i^{\text{adv}}))$$

$$\leq (1-\lambda)\tilde{O}(\eta\alpha^2)\phi\left(\alpha^3\sum_{r\in[m]}\left(u_r^{(t)}\right)^3\right) + \lambda\Theta\left(\frac{\eta\alpha^3(1-\gamma)^3}{N}\right)\sum_{i=1}^{N}\psi(y_i f_{\boldsymbol{W}^{(t)}}(\boldsymbol{X}_i^{\text{adv}})),$$

where we use $\phi(\cdot)$ to denote the logistics function defined as $\phi(z) = \log(1 + \exp(-z))$ and we derive the last inequality by Hypothesis E.1. Then, we know

$$\frac{1-\lambda}{N}\sum_{i=1}^{N}\psi(y_i f_{\boldsymbol{W}^{(t)}}(\boldsymbol{X}_i)) \leq (1-\lambda)\tilde{O}(\alpha)\phi\left(\alpha^3\sum_{r\in[m]}\left(u_r^{(t)}\right)^3\right)$$

$$+ \lambda\Theta\left(\frac{\alpha^2(1-\gamma)^3}{N}\right)\sum_{i=1}^{N}\psi(y_i f_{\boldsymbol{W}^{(t)}}(\boldsymbol{X}_i^{\text{adv}})).$$

Then, we derive the following result by Hypothesis E.1 and the above inequality.

$$
\begin{aligned}
\max_{r\in[m]} u_r^{(t+1)} &\leq \max_{r\in[m]} u_r^{(t)} + \frac{3\eta}{N}\left(\max_{r\in[m]} u_r^{(t)}\right)^2 \sum_{i=1}^{N}\left((1-\lambda)\alpha^3\psi(y_i f_{\boldsymbol{W}^{(t)}}(\boldsymbol{X}_i))\right.\\
&\quad \left. + \lambda\alpha^3(1-\gamma)^3\psi(y_i f_{\boldsymbol{W}^{(t)}}(\boldsymbol{X}_i^{\mathrm{adv}}))\right)\\
&\leq \max_{r\in[m]} u_r^{(t)} + \tilde{\Theta}(\eta\alpha)\frac{1-\lambda}{N}\sum_{i=1}^{N}\psi(y_i f_{\boldsymbol{W}^{(t)}}(\boldsymbol{X}_i)) + \tilde{\Theta}\left(\frac{\eta\alpha(1-\gamma)^3}{N}\right)\sum_{i=1}^{N}\psi(y_i f_{\boldsymbol{W}^{(t)}}(\boldsymbol{X}_i^{\mathrm{adv}}))\\
&\leq \max_{r\in[m]} u_r^{(t)} + (1-\lambda)\tilde{O}(\eta\alpha^2)\phi\left(\alpha^3\sum_{r\in[m]}\left(u_r^{(t)}\right)^3\right) + \tilde{\Theta}\left(\frac{\eta\alpha^3(1-\gamma)^3}{N}\right)\sum_{i=1}^{N}\psi(y_i f_{\boldsymbol{W}^{(t)}}(\boldsymbol{X}_i^{\mathrm{adv}}))\\
&\leq \max_{r\in[m]} u_r^{(t)} + \frac{(1-\lambda)\tilde{O}(\eta\alpha^2)}{1+\exp\left(\alpha^3\sum_{r\in[m]}\left(u_r^{(t)}\right)^3\right)} + \tilde{\Theta}\left(\frac{\eta\alpha^3(1-\gamma)^3}{N}\right)\sum_{i=1}^{N}\psi(y_i f_{\boldsymbol{W}^{(t)}}(\boldsymbol{X}_i^{\mathrm{adv}}))\\
&\leq \max_{r\in[m]} u_r^{(t)} + \frac{(1-\lambda)\tilde{O}(\eta\alpha^2)}{1+\exp\left(\tilde{\Omega}(1)\right)} + \tilde{\Theta}\left(\frac{\eta\alpha^3(1-\gamma)^3}{N}\right)\sum_{i=1}^{N}\psi(y_i f_{\boldsymbol{W}^{(t)}}(\boldsymbol{X}_i^{\mathrm{adv}})).
\end{aligned}
$$

By summing up iteration $s = T_0, \ldots, t-1$, we have the following result as

$$
\begin{aligned}
\max_{r\in[m]} u_r^{(t)} &\leq \max_{r\in[m]} u_r^{(T_0)} + \sum_{s=T_0}^{t-1}\frac{(1-\lambda)\tilde{O}(\eta\alpha^2)}{1+\exp\left(\tilde{\Omega}(1)\right)} + \left(\frac{\eta\alpha^3(1-\gamma)^3}{N}\right)\sum_{s=T_0}^{t-1}\sum_{i=1}^{N}\psi(y_i f_{\boldsymbol{W}^{(s)}}(\boldsymbol{X}_i^{\mathrm{adv}}))\\
&\leq \tilde{O}(\alpha^{-1}) + \tilde{O}\left(\frac{\eta\alpha^3(1-\gamma)^3}{N}\right)\sum_{s=T_0}^{t-1}\sum_{i=1}^{N}\psi\left(\alpha^3(1-\gamma)^3\sum_{k\in[m]}\left(u_k^{(s)}\right)^3 + \mathcal{V}_i^{(s)}\right).
\end{aligned}
$$

Therefore, we derive the conclusion of Theorem E.4. $\qquad\square$

Next, we prove the following theorem about the update of noise components.

**Lemma E.5.** *For each $r \in [m]$, $i \in [N]$ and $j \in [P] \setminus \mathrm{signal}(\boldsymbol{X}_i)$, any iteration $t_0, t$ such that $t_0 < t \leq T$, with high probability, it holds that*

$$
\left| v_{i,j,r}^{(t)} - v_{i,j,r}^{(t_0)} - \Theta\left(\frac{\eta\sigma^2 d}{N}\right)\sum_{s=t_0}^{t-1}\tilde{\psi}_i^{(s)}\left(v_{i,j,r}^{(s)}\right)^2 \right| \leq \tilde{O}\left(\frac{\lambda\eta\alpha^3(1-\gamma)^3}{N}\right)\sum_{s=t_0}^{t-1}\sum_{i=1}^{N}\psi(y_i f_{\boldsymbol{W}^{(s)}}(\boldsymbol{X}_i^{adv}))
$$
$$
+ \tilde{O}(P\sigma^2\alpha^{-1}\sqrt{d}),
$$

*where we use the notation $\tilde{\psi}_i^{(s)}$ to denote $(1-\lambda)\psi(y_i f_{\boldsymbol{W}^{(s)}}(\boldsymbol{X}_i)) + \lambda\psi(y_i f_{\boldsymbol{W}^{(s)}}(\boldsymbol{X}_i^{adv}))$.*

*Proof.* To obtain Lemma E.5, we prove the following stronger result by induction w.r.t. iteration $t$.

$$
\begin{aligned}
\left| v_{i,j,r}^{(t)} - v_{i,j,r}^{(t_0)} - \Theta\left(\frac{\eta\sigma^2 d}{N}\right)\sum_{s=t_0}^{t-1}\tilde{\psi}_i^{(s)}\left(v_{i,j,r}^{(s)}\right)^2 \right| &\leq \tilde{O}(P\sigma^2\alpha^{-1}\sqrt{d})\left(1+\lambda\alpha\eta+\frac{\alpha}{\sigma^2 d}\right)\sum_{q=0}^{t-t_0-1}(P^{-1}\sqrt{d})^{-q}\\
&\quad + \tilde{O}\left(\frac{\lambda\eta\alpha^3(1-\gamma)^3}{N}\right)\sum_{q=0}^{t-t_0-1}\sum_{s=t_0}^{t-q}\sum_{i=1}^{N}(P^{-1}\sqrt{d})^{-q}\psi(y_i f_{\boldsymbol{W}^{(s)}}(\boldsymbol{X}_i^{\mathrm{adv}}))
\end{aligned}
\tag{1}
$$

First, we project the training update on noise patch $\boldsymbol{X}_i[j]$ to verify the above inequality when $t = t_0 + 1$ as

$$\left| v_{i,j,r}^{(t_0+1)} - v_{i,j,r}^{(t_0)} - \Theta\left(\frac{\eta\sigma^2 d}{N}\right) \tilde{\psi}_i^{(t^0)} \left(v_{i,j,r}^{(s)}\right)^2 \right| \leq \Theta\left(\frac{\eta\sigma^2 d}{N}\right) \sum_{a=1}^{N} \sum_{b\neq\text{signal}(\boldsymbol{X}_a)} \tilde{\psi}_a^{(t_0)} \left(v_{a,b,r}^{(t_0)}\right)^2$$

$$\leq \Theta(\eta P\sigma^2\sqrt{d}) \frac{1-\lambda}{N} \sum_{i=1}^{N} \psi(y_i f_{\boldsymbol{W}^{(t_0)}}(\boldsymbol{X}_i))$$

$$+ \Theta(\eta P\sigma^2\sqrt{d}) \frac{\lambda}{N} \sum_{i=1}^{N} \psi(y_i f_{\boldsymbol{W}^{(t_0)}}(\boldsymbol{X}_i^{\text{adv}}))$$

$$\leq \tilde{O}(P\sigma^2\alpha^{-1}\sqrt{d}) \left(1 + \lambda\alpha\eta + \frac{\alpha}{\sigma^2 d}\right),$$

where we apply $\frac{1-\lambda}{N} \sum_{i=1}^{N} \psi(y_i f_{\boldsymbol{W}^{(t_0)}}(\boldsymbol{X}_i)) \leq \tilde{O}(\eta^{-1}\alpha^{-1}) \left(\max_{r\in[m]} u_r^{(t_0+1)} - \max_{r\in[m]} u_r^{(t_0)}\right) \leq \tilde{O}(\eta^{-1}\alpha^{-2})$ and $\sum_{i=1}^{N} \psi(y_i f_{\boldsymbol{W}^{(t_0)}}(\boldsymbol{X}_i^{\text{adv}})) \leq \tilde{O}(1)$ to derive the last inequality.

Next, we assume that the stronger result holds for iteration $t$, and then we prove the result for iteration $t+1$ as follow.

$$\left| v_{i,j,r}^{(t+1)} - v_{i,j,r}^{(t_0)} - \Theta\left(\frac{\eta\sigma^2 d}{N}\right) \sum_{s=t_0}^{t-1} \tilde{\psi}_i^{(s)} \left(v_{i,j,r}^{(s)}\right)^2 \right| \leq \Theta\left(\frac{\eta\sigma^2 d}{N}\right) \sum_{s=t_0}^{t-1} \sum_{a=1}^{N} \sum_{b\neq\text{signal}(\boldsymbol{X}_a)} \tilde{\psi}_a^{(s)} \left(v_{a,b,r}^{(s)}\right)^2$$

$$+ \Theta(\eta P\sigma^2\sqrt{d}) \frac{1-\lambda}{N} \sum_{i=1}^{N} \psi(y_i f_{\boldsymbol{W}^{(t)}}(\boldsymbol{X}_i)) + \Theta(\eta P\sigma^2\sqrt{d}) \frac{\lambda}{N} \sum_{i=1}^{N} \psi(y_i f_{\boldsymbol{W}^{(t)}}(\boldsymbol{X}_i^{\text{adv}})).$$

Then, we bound the first term in the right of the above inequality by our induction hypothesis for $t$, and we can derive

$$\left| v_{i,j,r}^{(t+1)} - v_{i,j,r}^{(t_0)} - \Theta\left(\frac{\eta\sigma^2 d}{N}\right) \sum_{s=t_0}^{t-1} \tilde{\psi}_i^{(s)} \left(v_{i,j,r}^{(s)}\right)^2 \right| \leq \tilde{O}(P\sigma^2\alpha^{-1}\sqrt{d}) \left(1 + \lambda\alpha\eta + \frac{\alpha}{\sigma^2 d}\right) \sum_{q=0}^{t-t_0-1} (P^{-1}\sqrt{d})^{-q}$$

$$+ \tilde{O}\left(\frac{\lambda\eta\alpha^3(1-\gamma)^3}{N}\right) \sum_{q=0}^{t-t_0-1} \sum_{s=t_0}^{t-q} \sum_{i=1}^{N} (P^{-1}\sqrt{d})^{-q} \psi(y_i f_{\boldsymbol{W}^{(s)}}(\boldsymbol{X}_i^{\text{adv}}))$$

$$+ \tilde{O}(P\sigma^2\alpha^{-1}\sqrt{d}) \left(1 + \lambda\alpha\eta + \frac{\alpha}{\sigma^2 d}\right) + \Theta(\eta P\sigma^2\sqrt{d}) \frac{\lambda}{N} \sum_{i=1}^{N} \psi(y_i f_{\boldsymbol{W}^{(t)}}(\boldsymbol{X}_i^{\text{adv}})).$$

By summing up the terms, we proved the stronger result for $t+1$.

Finally, we simplify the form of stronger result by using $\sum_{q=0}^{\infty}(P^{-1}\sqrt{d})^{-q} = (1 - P/\sqrt{d})^{-1} = \Theta(1)$, which implies the conclusion of Lemma E.5. □

Now, we prove Lemma 5.13 based on Lemma E.5 as follow.

**Theorem E.6.** *(Restatement of Lemma 5.13) For each $i \in [N]$, $r \in [m]$ and $j \in [P] \setminus \text{signal}(\boldsymbol{X}_i)$ and any $t \geq 1$, the signal component grows as*

$$v_{i,j,r}^{(t)} \geq v_{i,j,r}^{(0)} + \Theta\left(\frac{\eta\sigma^2 d}{N}\right) \sum_{s=0}^{t-1} \psi(\mathcal{V}_i^{(s)}) \left(v_{i,j,r}^{(s)}\right)^2 - \tilde{O}(P\sigma^2\alpha^{-1}\sqrt{d}).$$

*Proof.* By applying the one-side inequality of Lemma E.5, we have

$$v_{i,j,r}^{(t)} - v_{i,j,r}^{(t_0)} - \Theta\left(\frac{\eta\sigma^2 d}{N}\right) \sum_{s=t_0}^{t-1} \tilde{\psi}_i^{(s)} \left(v_{i,j,r}^{(s)}\right)^2 \geq -\tilde{O}\left(\frac{\lambda\eta\alpha^3(1-\gamma)^3}{N}\right) \sum_{s=t_0}^{t-1} \sum_{i=1}^{N} \psi(y_i f_{\boldsymbol{W}^{(s)}}(\boldsymbol{X}_i^{\text{adv}}))$$

$$- \tilde{O}(P\sigma^2\alpha^{-1}\sqrt{d}).$$

Thus, we obtain Theorem E.6 by using $\tilde{O}\left(\frac{\lambda\eta\alpha^3(1-\gamma)^3}{N}\right) \sum_{s=t_0}^{t-1} \sum_{i=1}^{N} \psi(y_i f_{\boldsymbol{W}^{(s)}}(\boldsymbol{X}_i^{\text{adv}})) \leq \tilde{O}(\lambda\eta T\alpha^3(1-\gamma)^3) \leq \tilde{O}(P\sigma^2\alpha^{-1}\sqrt{d})$ and $\tilde{\psi}_i^{(s)} = \Theta(1)\psi(\mathcal{V}_i^{(s)})$ derived by Hypothesis E.1. □

Consequently, we derive the upper bound of total noise components as follow.

**Lemma E.7.** *During adversarial training, with high probability, it holds that, after $T_1 = \Theta\left(\frac{N}{\eta\sigma_0\sigma^3 d^{\frac{3}{2}}}\right)$ iterations, for all $t \in [T_1, T]$ and each $i \in [N]$, we have $\mathcal{V}_i^{(t)} \geq \tilde{O}(1)$.*

*Proof.* By applying Lemma E.5 as the same in the proof of Theorem E.6, we know that

$$\left| v_{i,j,r}^{(t)} - v_{i,j,r}^{(t_0)} - \Theta\left(\frac{\eta\sigma^2 d}{N}\right) \sum_{s=t_0}^{t-1} \tilde{\psi}_i^{(s)} \left(v_{i,j,r}^{(s)}\right)^2 \right| \leq \tilde{O}(P\sigma^2\alpha^{-1}\sqrt{d}),$$

which implies that, for any iteration $t \leq T$, we have

$$\begin{cases} v_{i,j,r}^{(t)} \geq v_{i,j,r}^{(0)} + A \sum_{s=0}^{t-1} \left(v_{i,j,r}^{(s)}\right)^2 - C \\ v_{i,j,r}^{(t)} \leq v_{i,j,r}^{(0)} + A \sum_{s=0}^{t-1} \left(v_{i,j,r}^{(s)}\right)^2 + C \end{cases},$$

where $A, C > 0$ are constants defined as

$$A = \frac{\tilde{\Theta}\left(\eta\sigma^2 d\right)}{N}, \quad C = \tilde{O}(P\sigma^2\alpha^{-1}\sqrt{d}).$$

At initialization, since we choose the weights $\boldsymbol{w}_r^{(0)} \sim \mathcal{N}\left(0, \sigma_0^2 \mathbf{I}_d\right)$ and $\boldsymbol{X}_i[j] \sim \mathcal{N}\left(0, \sigma^2 \mathbf{I}_d\right)$, we know the initial noise components $v_{i,j,r}^{(0)}$ are i.i.d. zero-mean Gaussian random variables, which implies that, with high probability, there exists at least one index $r'$ such that $v_{i,j,r}^{(0)} \geq \tilde{\Omega}(P\sigma^2\alpha^{-1}\sqrt{d})$. By using Tensor Power Method (Lemma C.2) and setting $v = \tilde{\Theta}(1)$, we have the threshold iteration $T_1$ as

$$T_1 = \frac{21N}{\tilde{\Theta}\left(\eta\sigma^2 d\right) v_{i,j,r}^{(0)}} + \frac{8N}{\tilde{\Theta}\left(\eta\sigma^2 d\right)\left(v_{i,j,r}^{(0)}\right)} \left\lceil \frac{\log\left(\frac{\tilde{O}(1)}{v_{i,j,r}^{(0)}}\right)}{\log(2)} \right\rceil.$$

Therefore, we get $T_1 = \Theta\left(\frac{N}{\eta\sigma_0\sigma^3 d^{\frac{3}{2}}}\right)$, and we use $\mathcal{V}_i^{(t)} = \sum_{r\in[m]} \sum_{j\in\in[P]\backslash \text{signal}(\boldsymbol{X}_i)} \left(v_{i,j,r}^{(t)}\right)^3$ to derive $\mathcal{V}_i^{(t)} \geq \tilde{\Omega}(1)$. $\square$

Indeed, our aimed loss function $\widehat{\mathcal{L}}_{\text{adv}}(\boldsymbol{W})$ is non-convex due to the non-linearity of our CNN model $f_{\boldsymbol{W}}$. To analyze the convergence of gradient algorithm, we need to prove the following condition that is used to show non-convexly global convergence (Karimi et al., 2016; Li et al., 2019).

**Lemma E.8.** *(Lojasiewicz Inequality for Non-convex Optimization) During adversarial training, with high probability, it holds that, after $T_1 = \Theta\left(\frac{N}{\eta\sigma_0\sigma^3 d^{\frac{3}{2}}}\right)$ iterations, for all $t \in [T_1, T]$, we have*

$$\left\| \nabla_{\boldsymbol{W}} \widehat{\mathcal{L}}_{adv}\left(\boldsymbol{W}^{(t)}\right) \right\|_2 \geq \tilde{\Omega}(1)\widehat{\mathcal{L}}_{adv}\left(\boldsymbol{W}^{(t)}\right).$$

*Proof.* To prove Lojasiewicz Inequality, we first recall the gradient w.r.t. $\boldsymbol{w}_r$ as

$$\nabla_{\boldsymbol{w}_r} \widehat{\mathcal{L}}_{\text{adv}}(\boldsymbol{W}^{(t)}) = -\frac{3}{N}\left(\left(u_r^{(t)}\right)^2 \left(\sum_{i=1}^{N} (1-\lambda)\alpha^3 \psi(y_i f_{\boldsymbol{W}^{(t)}}(\boldsymbol{X}_i)) + \lambda\alpha^3(1-\gamma)^3 \psi(y_i f_{\boldsymbol{W}^{(t)}}(\boldsymbol{X}_i^{\text{adv}}))\right) \boldsymbol{w}^*\right.$$

$$\left. + \sum_{i=1}^{N} \sum_{j\neq\text{signal}(\boldsymbol{X}_i)} \left(v_{i,j,r}^{(t)}\right)^2 \left((1-\lambda)\psi(y_i f_{\boldsymbol{W}^{(t)}}(\boldsymbol{X}_i)) + \lambda\psi(y_i f_{\boldsymbol{W}^{(t)}}(\boldsymbol{X}_i^{\text{adv}}))\right) \boldsymbol{X}_i[j]\right).$$

Then, we project the gradient on the signal direction and total noise, respectively.

For the signal component, we have

$$\left\| \nabla_{\boldsymbol{w}_r} \widehat{\mathcal{L}}_{\text{adv}}(\boldsymbol{W}^{(t)}) \right\|_2^2 \geq \left\langle \nabla_{\boldsymbol{w}_r} \widehat{\mathcal{L}}_{\text{adv}}(\boldsymbol{W}^{(t)}), \boldsymbol{w}^* \right\rangle^2$$

$$\geq \tilde{\Omega}(1) \left( (1-\lambda)\alpha^3 \left( u_r^{(t)} \right)^2 \psi \left( \alpha^3 \sum_{k \in [m]} \left( u_k^{(t)} \right)^3 \right) \right)^2.$$

For the total noise component, we have

$$\left\| \nabla_{\boldsymbol{w}_r} \widehat{\mathcal{L}}_{\text{adv}}(\boldsymbol{W}^{(t)}) \right\|_2^2 \geq \left\langle \nabla_{\boldsymbol{w}_r} \widehat{\mathcal{L}}_{\text{adv}}(\boldsymbol{W}^{(t)}), \frac{\sum_{i=1}^N \sum_{j \neq \text{signal}(\boldsymbol{X}_i)} \boldsymbol{X}_i[j]}{\left\| \sum_{i=1}^N \sum_{j \neq \text{signal}(\boldsymbol{X}_i)} \boldsymbol{X}_i[j] \right\|_2} \right\rangle^2$$

$$= \left\langle -\frac{3}{N} \sum_{i=1}^N \sum_{j \neq \text{signal}(\boldsymbol{X}_i)} \left( v_{i,j,r}^{(t)} \right)^2 ((1-\lambda)\psi(y_i f_{\boldsymbol{W}^{(t)}}(\boldsymbol{X}_i)) + \lambda\psi(y_i f_{\boldsymbol{W}^{(t)}}(\boldsymbol{X}_i^{\text{adv}}))) \boldsymbol{X}_i[j], \right.$$

$$\left. \frac{\sum_{a=1}^N \sum_{b \neq \text{signal}(\boldsymbol{X}_a)} \boldsymbol{X}_a[b]}{\left\| \sum_{a=1}^N \sum_{b \neq \text{signal}(\boldsymbol{X}_a)} \boldsymbol{X}_a[b] \right\|_2} \right\rangle^2$$

$$= \left( \left\langle -\frac{3}{N} \sum_{i=1}^N \sum_{j \neq \text{signal}(\boldsymbol{X}_i)} \left( v_{i,j,r}^{(t)} \right)^2 (1-\lambda)\psi(y_i f_{\boldsymbol{W}^{(t)}}(\boldsymbol{X}_i))\boldsymbol{X}_i[j] , \frac{\sum_{a=1}^N \sum_{b \neq \text{signal}(\boldsymbol{X}_a)} \boldsymbol{X}_a[b]}{\left\| \sum_{a=1}^N \sum_{b \neq \text{signal}(\boldsymbol{X}_a)} \boldsymbol{X}_a[b] \right\|_2} \right\rangle \right.$$

$$\left. + \left\langle -\frac{3}{N} \sum_{i=1}^N \sum_{j \neq \text{signal}(\boldsymbol{X}_i)} \left( v_{i,j,r}^{(t)} \right)^2 \lambda\psi(y_i f_{\boldsymbol{W}^{(t)}}(\boldsymbol{X}_i^{\text{adv}}))\boldsymbol{X}_i[j], \frac{\sum_{a=1}^N \sum_{b \neq \text{signal}(\boldsymbol{X}_a)} \boldsymbol{X}_a[b]}{\left\| \sum_{a=1}^N \sum_{b \neq \text{signal}(\boldsymbol{X}_a)} \boldsymbol{X}_a[b] \right\|_2} \right\rangle \right)^2.$$

For the first term, with high probability, it holds that

$$\left\langle -\frac{3}{N} \sum_{i=1}^N \sum_{j \neq \text{signal}(\boldsymbol{X}_i)} \left( v_{i,j,r}^{(t)} \right)^2 (1-\lambda)\psi(y_i f_{\boldsymbol{W}^{(t)}}(\boldsymbol{X}_i))\boldsymbol{X}_i[j], \frac{\sum_{a=1}^N \sum_{b \neq \text{signal}(\boldsymbol{X}_a)} \boldsymbol{X}_a[b]}{\left\| \sum_{a=1}^N \sum_{b \neq \text{signal}(\boldsymbol{X}_a)} \boldsymbol{X}_a[b] \right\|_2} \right\rangle$$

$$\geq -\frac{\tilde{O}(\sigma)(1-\lambda)}{N} \sum_{i=1}^N \sum_{j \neq \text{signal}(\boldsymbol{X}_i)} \left( v_{i,j,r}^{(t)} \right)^2 (1-\lambda)\psi(y_i f_{\boldsymbol{W}^{(t)}}(\boldsymbol{X}_i)),$$

where we use that $\left\langle \boldsymbol{X}_i[j], \frac{\sum_{a=1}^N \sum_{b \neq \text{signal}(\boldsymbol{X}_a)} \boldsymbol{X}_a[b]}{\left\| \sum_{a=1}^N \sum_{b \neq \text{signal}(\boldsymbol{X}_a)} \boldsymbol{X}_a[b] \right\|_2} \right\rangle$ is a sub-Gaussian random variable of parameter $\sigma$, which implies

w.h.p. $\left| \left\langle \boldsymbol{X}_i[j], \frac{\sum_{a=1}^N \sum_{b \neq \text{signal}(\boldsymbol{X}_a)} \boldsymbol{X}_a[b]}{\left\| \sum_{a=1}^N \sum_{b \neq \text{signal}(\boldsymbol{X}_a)} \boldsymbol{X}_a[b] \right\|_2} \right\rangle \right| \leq \tilde{O}(\sigma).$

For the second term, with high probability, it holds that

$$\left\langle \frac{3}{N} \sum_{i=1}^N \sum_{j \neq \text{signal}(\boldsymbol{X}_i)} \left( v_{i,j,r}^{(t)} \right)^2 \lambda\psi(y_i f_{\boldsymbol{W}^{(t)}}(\boldsymbol{X}_i^{\text{adv}}))\boldsymbol{X}_i[j], \frac{\sum_{a=1}^N \sum_{b \neq \text{signal}(\boldsymbol{X}_a)} \boldsymbol{X}_a[b]}{\left\| \sum_{a=1}^N \sum_{b \neq \text{signal}(\boldsymbol{X}_a)} \boldsymbol{X}_a[b] \right\|_2} \right\rangle$$

$$= \frac{\Theta(1)}{N} \sum_{i=1}^N \sum_{j \neq \text{signal}(\boldsymbol{X}_i)} \left( v_{i,j,r}^{(t)} \right)^2 \lambda\psi(y_i f_{\boldsymbol{W}^{(t)}}(\boldsymbol{X}_i^{\text{adv}})) \frac{\|\boldsymbol{X}_i[j]\|_2^2}{\left\| \sum_{a=1}^N \sum_{b \neq \text{signal}(\boldsymbol{X}_a)} \boldsymbol{X}_a[b] \right\|_2}$$

$$= \frac{\Theta(\sigma\sqrt{d})}{N} \sum_{i=1}^N \sum_{j \neq \text{signal}(\boldsymbol{X}_i)} \left( v_{i,j,r}^{(t)} \right)^2 \lambda\psi(y_i f_{\boldsymbol{W}^{(t)}}(\boldsymbol{X}_i^{\text{adv}})),$$

where we use w.h.p. $\dfrac{\langle \boldsymbol{X}_i[j], \boldsymbol{X}_{i'}[j'] \rangle}{\left\| \sum_{a=1}^{N} \sum_{b \neq \text{signal}(\boldsymbol{X}_a)} \boldsymbol{X}_a[b] \right\|_2} \leq \dfrac{\Theta\left(\frac{1}{\sqrt{d}}\right) \|\boldsymbol{X}_i[j]\|_2^2}{\left\| \sum_{a=1}^{N} \sum_{b \neq \text{signal}(\boldsymbol{X}_a)} \boldsymbol{X}_a[b] \right\|_2}$ for $(i,j) \neq (i',j')$.

Now, combine the above bounds, we derive

$$
\begin{aligned}
\sum_{r=1}^{m} \left\| \nabla_{\boldsymbol{w}_r} \widehat{\mathcal{L}}_{\text{adv}}(\boldsymbol{W}^{(t)}) \right\|_2^2 &\geq \sum_{r=1}^{m} \left\langle \nabla_{\boldsymbol{w}_r} \widehat{\mathcal{L}}_{\text{adv}}(\boldsymbol{W}^{(t)}), \boldsymbol{w}^* \right\rangle^2 \\
&\quad + \sum_{r=1}^{m} \left\langle \nabla_{\boldsymbol{w}_r} \widehat{\mathcal{L}}_{\text{adv}}(\boldsymbol{W}^{(t)}), \frac{\sum_{i=1}^{N} \sum_{j \neq \text{signal}(\boldsymbol{X}_i)} \boldsymbol{X}_i[j]}{\left\| \sum_{i=1}^{N} \sum_{j \neq \text{signal}(\boldsymbol{X}_i)} \boldsymbol{X}_i[j] \right\|_2} \right\rangle^2 \\
&\geq \Omega\left(\frac{1}{m}\right) \left( (1-\lambda)\alpha^3 \sum_{r=1}^{m} \left(u_r^{(t)}\right)^2 \psi\left( \alpha^3 \sum_{k \in [m]} \left(u_k^{(t)}\right)^3 \right) \right. \\
&\quad + \frac{\Theta(\sigma\sqrt{d})}{N} \sum_{r=1}^{m} \sum_{i=1}^{N} \sum_{j \neq \text{signal}(\boldsymbol{X}_i)} \left(v_{i,j,r}^{(t)}\right)^2 \lambda \psi(y_i f_{\boldsymbol{W}^{(t)}}(\boldsymbol{X}_i^{\text{adv}})) \\
&\quad \left. - \frac{\tilde{O}(\sigma)(1-\lambda)}{N} \sum_{r=1}^{m} \sum_{i=1}^{N} \sum_{j \neq \text{signal}(\boldsymbol{X}_i)} \left(v_{i,j,r}^{(t)}\right)^2 (1-\lambda) \psi(y_i f_{\boldsymbol{W}^{(t)}}(\boldsymbol{X}_i)) \right)^2 \\
&\geq \tilde{\Omega}(1) \left( (1-\lambda)\phi\left( \alpha^3 \sum_{r \in [m]} \left(u_r^{(t)}\right)^3 \right) + \frac{\lambda}{N} \sum_{i=1}^{N} \phi\left( \mathcal{V}_i^{(t)} \right) \right)^2 \\
&\geq \tilde{\Omega}(1) \left( \widehat{\mathcal{L}}_{\text{adv}}\left( \boldsymbol{W}^{(t)} \right) \right)^2.
\end{aligned}
$$

$\square$

Consequently, we derive the following sub-linear convergence result by applying Lojasiewicz Inequality.

**Lemma E.9.** *(Sub-linear Convergence for Adversarial Training) During adversarial training, with high probability, it holds that, after $T_1 = \Theta\left( \frac{N}{\eta \sigma_0 \sigma^3 d^{\frac{3}{2}}} \right)$ iterations, the adversarial training loss sub-linearly converges to zero as*

$$
\widehat{\mathcal{L}}_{adv}\left( \boldsymbol{W}^{(t)} \right) \leq \frac{\tilde{O}(1)}{\eta(t - T_1 + 1)}.
$$

*Proof.* Due to the smoothness of loss function $\widehat{\mathcal{L}}_{\text{adv}}(\boldsymbol{W})$ and learning rate $\eta = \tilde{O}(1)$, we have

$$
\begin{aligned}
\widehat{\mathcal{L}}_{\text{adv}}\left( \boldsymbol{W}^{(t+1)} \right) &\leq \widehat{\mathcal{L}}_{\text{adv}}\left( \boldsymbol{W}^{(t)} \right) - \frac{\eta}{2} \left\| \nabla_{\boldsymbol{W}} \widehat{\mathcal{L}}_{\text{adv}}\left( \boldsymbol{W}^{(t)} \right) \right\|_2 \\
&\leq \widehat{\mathcal{L}}_{\text{adv}}\left( \boldsymbol{W}^{(t)} \right) - \tilde{\Omega}(\eta) \left( \widehat{\mathcal{L}}_{\text{adv}}\left( \boldsymbol{W}^{(t)} \right) \right)^2,
\end{aligned}
$$

where we use Lojasiewicz Inequality in the last inequality. Then, by applying Tensor Power Method (Lemma C.3), we obtain the sub-linear convergence rate. $\square$

Now, we present the following result to bound the derivative generated by training-adversarial examples.

**Lemma E.10.** *During adversarial training, with high probability, it holds that, after $T_1 = \Theta\left( \frac{N}{\eta \sigma_0 \sigma^3 d^{\frac{3}{2}}} \right)$ iterations, we have $\frac{\lambda}{N} \sum_{s=0}^{t} \sum_{i=1}^{N} \psi(y_i f_{\boldsymbol{W}^{(s)}}(\boldsymbol{X}_i^{adv})) \leq \tilde{O}(\eta^{-1} \sigma_0^{-1}).$*

*Proof.* First, we bound the total derivative during iteration $s = T_1, \ldots, t$. By applying the conclusion of Lemma E.5, we

have

$$\frac{\lambda}{N} \sum_{s=T_1}^{t} \sum_{i=1}^{N} \psi(y_i f_{\boldsymbol{W}^{(s)}}(\boldsymbol{X}_i^{\text{adv}})) \leq \frac{\tilde{O}(1)}{N} \sum_{s=T_1}^{t-1} \sum_{i=1}^{N} \tilde{\psi}_i^{(s)} \left(v_{i,j,r}^{(s)}\right)^2$$
$$+ \tilde{O}\left(\frac{\lambda \alpha^3 (1-\gamma)^3}{N\sigma^2 d}\right) \sum_{s=T_1}^{t-1} \sum_{i=1}^{N} \psi(y_i f_{\boldsymbol{W}^{(s)}}(\boldsymbol{X}_i^{\text{adv}})) + \tilde{O}\left(\frac{P}{\eta \alpha \sqrt{d}}\right).$$

Due to $\tilde{O}\left(\frac{\alpha^3(1-\gamma)^3}{\sigma^2 d}\right) \ll 1$, we know

$$\frac{\lambda}{N} \sum_{s=T_1}^{t} \sum_{i=1}^{N} \psi(y_i f_{\boldsymbol{W}^{(s)}}(\boldsymbol{X}_i^{\text{adv}})) \leq \frac{\tilde{O}(1)}{N} \sum_{s=T_1}^{t-1} \sum_{i=1}^{N} \tilde{\psi}_i^{(s)} \left(v_{i,j,r}^{(s)}\right)^2 + \tilde{O}\left(\frac{P}{\eta \alpha \sqrt{d}}\right)$$
$$\leq \frac{\tilde{O}(1)}{N} \sum_{s=T_1}^{t-1} \sum_{i=1}^{N} \phi\left(\mathcal{V}_i^{(s)}\right) + \tilde{O}\left(\frac{P}{\eta \alpha \sqrt{d}}\right)$$
$$\leq \tilde{O}(1) \sum_{s=T_1}^{t-1} \widehat{\mathcal{L}}_{\text{adv}}\left(\boldsymbol{W}^{(t)}\right) + \tilde{O}\left(\frac{P}{\eta \alpha \sqrt{d}}\right)$$
$$\leq \tilde{O}(1) \sum_{s=T_1}^{t-1} \frac{\tilde{O}(1)}{\eta(t - T_1 + 1)} + \tilde{O}\left(\frac{P}{\eta \alpha \sqrt{d}}\right) \leq \tilde{O}(\eta^{-1}).$$

Thus, we obtain $\frac{\lambda}{N} \sum_{s=0}^{t} \sum_{i=1}^{N} \psi(y_i f_{\boldsymbol{W}^{(s)}}(\boldsymbol{X}_i^{\text{adv}})) \leq \tilde{O}(\sigma_0^{-1}) + \tilde{O}(\eta^{-1}) \leq \tilde{O}(\eta^{-1}\sigma_0^{-1}).$ □

Consequently, we have the following lemma that verifies Hypothesis E.1 for $t = T$.

**Lemma E.11.** *During adversarial training, with high probability, it holds that, for any $t \leq T$, we have $\max_{r \in [m]} u_r^{(t)} \leq \tilde{O}(\alpha^{-1})$ and $|v_{i,j,r}^{(t)}| \leq \tilde{O}(1)$ for each $r \in [m], i \in [N], j \in [P] \setminus \text{signal}(\boldsymbol{X}_i)$.*

*Proof.* Combined with Theorem E.4 and Lemma E.10, we can derive $\max_{r \in [m]} u_r^{(T)} \leq \tilde{O}(\alpha^{-1})$.

By applying Lemma E.5, we have

$$|v_{i,j,r}^{(T)}| \leq |v_{i,j,r}^{(0)}| + \Theta\left(\frac{\eta \sigma^2 d}{N}\right) \sum_{s=t_0}^{t-1} \tilde{\psi}_i^{(s)} \left(v_{i,j,r}^{(s)}\right)^2$$
$$+ \tilde{O}\left(\frac{\lambda \eta \alpha^3 (1-\gamma)^3}{N}\right) \sum_{s=t_0}^{t-1} \sum_{i=1}^{N} \psi(y_i f_{\boldsymbol{W}^{(s)}}(\boldsymbol{X}_i^{\text{adv}})) + \tilde{O}(P\sigma^2 \alpha^{-1}\sqrt{d})$$
$$\leq \tilde{O}(1) + \tilde{O}(\sigma^2 d) + \tilde{O}(\alpha^3(1-\gamma)^3 \sigma_0^{-1}) + \tilde{O}(P\sigma^2 \alpha^{-1}\sqrt{d}) \leq \tilde{O}(1).$$

Therefore, our Hypothesis E.1 holds for iteration $t = T$. □

Finally, we prove our main result as follow.

**Theorem E.12.** *(Restatement of Theorem 5.9) Under Assumption 5.7, we run the adversarial training algorithm to update the weight of the simplified CNN model for $T = \Omega(\text{poly}(d))$ iterations. Then, with high probability, it holds that the CNN model*

1. *partially learns the true feature, i.e. $\mathcal{U}^{(T)} = \Theta(\alpha^{-3})$;*

2. *exactly memorizes the spurious feature, i.e. for each $i \in [N], \mathcal{V}_i^{(T)} = \Theta(1)$,*

*where $\mathcal{U}^{(t)}$ and $\mathcal{V}_i^{(t)}$ is defined for $i-$th instance $(\boldsymbol{X}_i, y_i)$ and $t-$th iteration as the same in (1)(1). Consequently, the clean test error and robust training error are both smaller than $o(1)$, but the robust test error is at least $\frac{1}{2} - o(1)$.*

*Proof.* First, by applying Lemma E.3, Lemma E.7 and Lemma E.11, we know for any $i \in [N]$

$$\mathcal{U}^{(T)} = \sum_{r \in [m]} \left( u_r^{(T)} \right)^3 = \Theta(\alpha^{-3})$$

$$\mathcal{V}_i^{(T)} = \sum_{r \in [m]} \sum_{j \neq \text{signal}(\boldsymbol{X}_i)} \left( v_{i,j,r}^{(T)} \right)^3 = \Theta(1).$$

Then, since adversarial loss sub-linearly converges to zero i.e. $\widehat{\mathcal{L}}_{\text{adv}} \left( \boldsymbol{W}^{(T)} \right) \leq \frac{\tilde{O}(1)}{\eta(T - T_1 + 1)} \leq \tilde{O} \left( \frac{1}{\text{poly}(d)} \right) = o(1)$, the robust training error is also at most $o(1)$.

To analyze test errors, we decompose $\boldsymbol{w}_r^{(T)}$ into $\boldsymbol{w}_r^{(T)} = \mu_r^{(T)} \boldsymbol{w}^* + \boldsymbol{\beta}_r^{(T)}$ for each $r \in [m]$, where $\boldsymbol{\beta}_r^{(T)} \in (\text{span}(\boldsymbol{w}^*))^\perp$. Due to $\mathcal{V}_i^{(T)} = \Theta(1)$, we know $\|\boldsymbol{\beta}_r^{(T)}\|_2 = \Theta(1)$.

For the clean test error, we have

$$\mathbb{P}_{(\boldsymbol{X},y) \sim \mathcal{D}} \left[ y f_{\boldsymbol{W}^{(T)}}(\boldsymbol{X}) < 0 \right] = \mathbb{P}_{(\boldsymbol{X},y) \sim \mathcal{D}} \left[ \alpha^3 \sum_{r=1}^m \left( u_r^{(T)} \right)^3 + y \sum_{r=1}^m \sum_{j \in [P] \backslash \text{signal}(\boldsymbol{X})} \left\langle \boldsymbol{w}_r^{(T)}, \boldsymbol{X}[j] \right\rangle^3 < 0 \right]$$

$$\leq \mathbb{P}_{(\boldsymbol{X},y) \sim \mathcal{D}} \left[ \sum_{r=1}^m \sum_{j \in [P] \backslash \text{signal}(\boldsymbol{X})} \left\langle \boldsymbol{\beta}_r^{(T)}, \boldsymbol{X}[j] \right\rangle^3 \geq \tilde{\Omega}(1) \right]$$

$$\leq \exp \left( -\frac{\tilde{\Omega}(1)}{\sigma^6 \sum_{r=1}^m \|\boldsymbol{\beta}_r^{(T)}\|_2^6} \right) \leq O \left( \frac{1}{\text{poly}(d)} \right) = o(1),$$

where we use the fact that $\sum_{r=1}^m \sum_{j \in [P] \backslash \text{signal}(\boldsymbol{X})} \left\langle \boldsymbol{\beta}_r^{(T)}, \boldsymbol{X}[j] \right\rangle^3$ is a sub-Gaussian random variable with parameter $\sigma^3 \sqrt{(P-1) \sum_{r=1}^m \|\boldsymbol{\beta}_r^{(T)}\|_2^6}$.

For the robust test error, we use $\mathcal{A}(\cdot)$ to denote attack in Definition 5.6, and then we derive

$$\mathbb{P}_{(\boldsymbol{X},y) \sim \mathcal{D}} \left[ \min_{\|\boldsymbol{\xi}\|_2 \leq \delta} y f_{\boldsymbol{W}^{(T)}}(\boldsymbol{X} + \boldsymbol{\xi}) < 0 \right] \geq \mathbb{P}_{(\boldsymbol{X},y) \sim \mathcal{D}} \left[ y f_{\boldsymbol{W}^{(T)}}(\mathcal{A}(\boldsymbol{X})) < 0 \right]$$

$$= \mathbb{P}_{(\boldsymbol{X},y) \sim \mathcal{D}} \left[ \alpha^3 \sum_{r=1}^m \left( u_r^{(T)} \right)^3 (1 - \gamma)^3 + y \sum_{r=1}^m \sum_{j \in [P] \backslash \text{signal}(\boldsymbol{X})} \left\langle \boldsymbol{w}_r^{(T)}, \boldsymbol{X}[j] \right\rangle^3 < 0 \right]$$

$$\geq \frac{1}{2} \mathbb{P}_{(\boldsymbol{X},y) \sim \mathcal{D}} \left[ \left| \sum_{r=1}^m \sum_{j \in [P] \backslash \text{signal}(\boldsymbol{X})} \left\langle \boldsymbol{\beta}_r^{(T)}, \boldsymbol{X}[j] \right\rangle^3 \right| \geq \tilde{\Omega} \left( (1 - \gamma)^3 \right) \right] \geq \frac{1}{2} \left( 1 - \frac{\tilde{O}(d)}{2^d} \right) = \frac{1}{2} - o(1),$$

where we use Lemma C.4 in the last inequality. $\qquad \square$

# F. Proof for Section 4

We prove Theorem 4.4 by using ReLU network to approximate $f_S$ proposed in Section 1.

**Theorem F.1.** *(Restatement of Theorem 4.4) Under Assumption 4.1, 4.2 and 4.3, with $N-$sample training dataset $S = \{(\boldsymbol{X}_1, y_1), (\boldsymbol{X}_2, y_2), \ldots, (\boldsymbol{X}_N, y_N)\}$ drawn from the data distribution $\mathcal{D}$, there exists a CGRO classifier that can be represented as a ReLU network with $\mathrm{poly}(D) + \tilde{O}(ND)$ parameters, which means that, under the distribution $\mathcal{D}$ and dataset $S$, the network achieves zero clean test and robust training errors but its robust test error is at least $\Omega(1)$.*

*Proof.* First, we give the following useful results about function approximation by ReLU nets.

**Lemma F.2.** *(Yarotsky, 2017) The function $f(x) = x^2$ on the segment $[0, 1]$ can be approximated with any error $\epsilon > 0$ by a ReLU network having the depth and the number of weights and computation units $O(\log(1/\epsilon))$.*

**Lemma F.3.** *(Yarotsky, 2017) Let $\epsilon > 0$, $0 < a < b$ and $B \geq 1$ be given. There exists a function $\widetilde{\times} : [0, B]^2 \to [0, B^2]$ computed by a ReLU network with $O\left(\log^2\left(\epsilon^{-1} B\right)\right)$ parameters such that*

$$\sup_{x,y \in [0,B]} \left|\widetilde{\times}(x, y) - xy\right| \leq \epsilon,$$

*and $\widetilde{\times}(x, y) = 0$ if $xy = 0$.*

Since for $\forall \boldsymbol{X}_0 \in [0, 1]^D$, the $\ell_2-$distance function $\|\boldsymbol{X} - \boldsymbol{X}_0\|^2 = \sum_{i=1}^{D} |\boldsymbol{X}^{(i)} - \boldsymbol{X}_0^{(i)}|^2$, by using Lemma F.2, there exists a function $\phi_1$ computed by a ReLU network with $\mathcal{O}\left(D \log\left(\varepsilon_1^{-1} D\right)\right)$ parameters such that $\sup_{\boldsymbol{X} \in [0,1]^D} \left|\phi_1(\boldsymbol{X}) - \|\boldsymbol{X} - \boldsymbol{X}_0\|^2\right| \leq \varepsilon_1$.

Return to our main proof back, indeed, functions computed by ReLU networks are piecewise linear but the indicator functions are not continuous, so we need to relax the indicator such that $\hat{I}_{\mathrm{soft}}(x) = 1$ for $x \leq \delta + \epsilon_0$, $\hat{I}_{\mathrm{soft}}(x) = 0$ for $x \geq R - \delta\epsilon_0$ and $\hat{I}_{\mathrm{soft}}$ is linear in $(\delta + \epsilon_0, R - \delta\epsilon_0)$ by using only two ReLU neurons, where $\epsilon_0$ is sufficient small for approximation.

Now, we notice that the constructed function $f_S$ can be re-written as

$$f_S(\boldsymbol{X}) = f_{\mathrm{clean}}(\boldsymbol{X})\left(1 - \mathbb{I}\{\boldsymbol{X} \in \cup_{i=1}^N \mathbb{B}_2(\boldsymbol{X}_i, \delta)\}\right) + \sum_{i=1}^N y_i \mathbb{I}\{\boldsymbol{X} \in \mathbb{B}_2(\boldsymbol{X}_i, \delta)\}$$

$$= f_{\mathrm{clean}}(\boldsymbol{X}) + \sum_{i=1}^N (y_i - f_{\mathrm{clean}}(\boldsymbol{X}))\mathbb{I}\{\|\boldsymbol{X} - \boldsymbol{X_i}\|_2^2 \leq \delta^2\}.$$

Combined with Lemma F.2, Lemma F.3 and the relaxed indicator, we know that there exists a ReLU net $h$ with at most $\mathrm{poly}(D) + \tilde{O}(ND)$ parameters such that $|h - f_S| = o(1)$ for all input $X \in [0, 1]^D$. Thus, it is easy to check that $h$ belongs to CGRO classifiers. $\square$

Next, we prove Theorem 4.7 by using the VC-dimension theory.

**Theorem F.4.** *(Restatement of Theorem 4.7) Let $\mathcal{F}_M$ be the family of function represented by ReLU networks with at most $M$ parameters. There exists a number $M_D = \Omega(\exp(D))$ and a distribution $\mathcal{D}$ satisfying Assumption 4.1, 4.2 and 4.3 such that, for any classifier in the family $\mathcal{F}_{M_D}$, under the distribution $\mathcal{D}$, the robust test error is at least $\Omega(1)$.*

*Proof.* Now, we notice that ReLU networks are piece-wise linear functions. Montufar et al. (2014) study the number of local linear regions, which provides the following result.

**Proposition F.5.** *The maximal number of linear regions of the functions computed by any ReLU network with a total of $n$ hidden units is bounded from above by $2^n$.*

Thus, for a given clean classifier $f_{\mathrm{clean}}$ represented by a ReLU net with $\mathrm{poly}(D)$ parameters, we know there exists at least a local region $V$ such that decision boundary of $f_{\mathrm{clean}}$ is linear hyperplane in $V$. And we assume that the hyperplane is $\boldsymbol{X}^{(D)} = \frac{1}{2}$.

Then, let $V'$ be the projection of $V$ on the decision boundary of $f_{\text{clean}}$, and $\mathcal{P}$ be an $2\delta$-packing of $V'$. Since the packing number $\mathcal{P}(V', \|\cdot\|, 2\delta) \geq \mathcal{C}(V', \|\cdot\|_2, 2\delta) = \exp(\Omega(D))$, where $\mathcal{C}(\Theta, \|\cdot\|, \delta)$ is the $\delta$-covering number of a set $\Theta$. For any $\epsilon_0 \in (0,1)$, we can consider the construction

$$S_\phi = \left\{ \left( \boldsymbol{x}, \frac{1}{2} + \epsilon_0 \cdot \phi(\boldsymbol{x}) \right) : \boldsymbol{x} \in \mathcal{P} \right\},$$

where $\phi : \mathcal{P} \to \{-1, +1\}$ is an arbitrary mapping. It's easy to see that all points in $S_\phi$ with first $D-1$ components satisfying $\|\boldsymbol{x}\|_2 \leq \sqrt{1 - \epsilon_0^2}$ are in $V'$, so that by choosing $\epsilon_0$ sufficiently small, we can guarantee that $|S_\phi \cap V| = \exp(\Omega(D))$. For convenience we just replace $S_\phi$ with $S_\phi \cap V$ from now on.

Let $A_\phi = S_\phi \cap \{ \boldsymbol{X} \in V : \boldsymbol{x}^{(D)} > \frac{1}{2} \}$, $B_\phi = S_\phi - A_\phi$. It's easy to see that for arbitrary $\phi$, the construction is linear-separable and satisfies $2\delta$-separability.

Assume that for any choices of $\phi$, the induced sets $A_\phi$ and $B_\phi$ can always be robustly classified with $(O(\delta), 1 - \mu)$-accuracy by a ReLU network with at most $M$ parameters. Then, we can construct an *enveloping network* $F_\theta$ with $M - 1$ hidden layers, $M$ neurons per layer and at most $M^3$ parameters such that any network with size $\leq M$ can be embedded into this envelope network. As a result, $F_\theta$ is capable of $(O(\delta), 1 - \mu)$-robustly classify any sets $A_\phi, B_\phi$ induced by arbitrary choices of $\phi$. We use $R_\phi$ to denote the subset of $S_\phi = A_\phi \cup B_\phi$ satisfying $|R_\phi| = (1 - \mu)|S_\phi| = \exp(\Omega(D))$ such that $R_\phi$ can be $O(\delta)$-robustly classified.

Next, we estimate the lower and upper bounds for the cardinal number of the vector set

$$R := \{ (f(\boldsymbol{x}))_{\boldsymbol{x} \in \mathcal{P}} | f \in \mathcal{F}_{M_D} \}.$$

Let $n$ denote $|\mathcal{P}|$, then we have
$$R = \{ (f(\boldsymbol{x}_1), f(\boldsymbol{x}_2), ... f(\boldsymbol{x}_n)) | f \in \mathcal{F}_{M_D} \},$$
where $\mathcal{P} = \{ \boldsymbol{x}_1, \boldsymbol{x}_2, ..., \boldsymbol{x}_n \}$.

On one hand, we know that for any $u \in \{-1, 1\}^n$, there exists a $v \in R$ such that $d_H(u, v) \leq \alpha n$, where $d_H(\cdot, \cdot)$ denotes the Hamming distance, then we have

$$|R| \geq \mathcal{N}(\{-1, 1\}^n, d_H, \mu n) \geq \frac{2^n}{\sum_{i=0}^{\mu n} \binom{n}{i}}.$$

On the other hand, by applying Lemma C.8, we have

$$\frac{2^n}{\sum_{i=1}^{\mu n} \binom{n}{i}} \leq |R| \leq \Pi_{\mathcal{F}_{M_D}}(n) \leq \sum_{j=0}^{l} \binom{n}{j}.$$

where $l$ is the VC-dimension of $\mathcal{F}_{M_D}$. In fact, we can derive $l = \Omega(n)$ when $\mu$ is a small constant. Assume that $l < n - 1$, then we have $\sum_{j=0}^{l} \binom{n}{j} \leq (en/l)^l$ and $\sum_{i=1}^{\mu n} \binom{n}{i} \leq (e/\mu)^{\mu n}$, so

$$\frac{2^n}{(e/\mu)^{\mu n}} \leq |R| \leq (en/l)^l.$$

We define a function $h(x)$ as $h(x) = (e/x)^x$, then we derive

$$2 \leq \left( \frac{e}{\mu} \right)^{\mu} \left( \frac{e}{l/n} \right)^{l/n} = h(\mu)h(l/n).$$

When $\mu$ is sufficient small, $l/n \geq C(\mu)$ that is a constant only depending on $\mu$, which implies $l = \Omega(n)$. Finally, by using Lemma C.7 and $n = |\mathcal{P}| = \exp(\Omega(D))$, we know $M_D = \exp(\Omega(D))$. $\qquad \square$

## G. Proof for Section B

**Theorem G.1.** *(Restatement of Theorem B.1) Let $\mathcal{D}$ be the underlying distribution with a smooth density function, and $N-$sample training dataset $\mathcal{S} = \{(\boldsymbol{X}_1, y_1), (\boldsymbol{X}_2, y_2), \ldots, (\boldsymbol{X}_N, y_N)\}$ is i.i.d. drawn from $\mathcal{D}$. Then, with high probability, it holds that,*

$$\mathcal{L}_{adv}(f) \leq \widehat{\mathcal{L}}_{adv}(f) + N^{-\frac{1}{D+2}} O\left( \underbrace{\mathbb{E}_{(\boldsymbol{X},y)\sim\mathcal{D}}\left[ \max_{\|\boldsymbol{\xi}\|_p \leq \delta} \|\nabla_{\boldsymbol{X}} \mathcal{L}(f(\boldsymbol{X}+\boldsymbol{\xi}), y)\|_q \right]}_{\text{global flatness}} \right).$$

*Proof.* Indeed, we notice the following loss decomposition,

$$\mathcal{L}_{\text{adv}}(f) - \widehat{\mathcal{L}}_{\text{adv}}(f) = \left( \mathcal{L}_{\text{clean}}(f) - \widehat{\mathcal{L}}_{\text{adv}}(f) \right) + \left( \mathcal{L}_{\text{adv}}(f) - \mathcal{L}_{\text{clean}}(f) \right).$$

To bound the first term, by applying $\lambda_i$ to denote kernel density estimation (KDE) proposed in Petzka et al. (2020), then we derive

$$
\begin{aligned}
\mathcal{L}_{\text{clean}}(f) - \widehat{\mathcal{L}}_{\text{adv}}(f) &= \mathbb{E}_{(\boldsymbol{X},y)\sim\mathcal{D}}[\mathcal{L}(f(\boldsymbol{X}), y)] - \frac{1}{N}\sum_{i=1}^{N} \max_{\|\boldsymbol{\xi}\|_p \leq \delta} \mathcal{L}(f(\boldsymbol{X}_i + \boldsymbol{\xi}, y_i)) \\
&\leq \mathbb{E}_{(\boldsymbol{X},y)\sim\mathcal{D}}[\mathcal{L}(f(\boldsymbol{X}), y)] - \frac{1}{N}\sum_{i=1}^{N} \mathbb{E}_{\boldsymbol{\xi}\sim\lambda_i}[\mathcal{L}(f(\boldsymbol{X}_i + \boldsymbol{\xi}), y_i)] \\
&= \int_{\boldsymbol{X}} p_{\mathcal{D}}(\boldsymbol{X})\mathcal{L}(f(\boldsymbol{X}), y)d\boldsymbol{X} - \int_{\boldsymbol{X}} p_{\mathcal{S}}(\boldsymbol{X})\mathcal{L}(f(\boldsymbol{X}), y)d\boldsymbol{X} \\
&\leq \underbrace{\left| \int_{\boldsymbol{X}} (p_{\mathcal{D}}(\boldsymbol{X}) - \mathbb{E}_{\mathcal{S}}[p_{\mathcal{S}}(\boldsymbol{X})])\mathcal{L}(f(\boldsymbol{X}), y(\boldsymbol{X}))d\boldsymbol{X} \right|}_{(I)} \\
&+ \underbrace{\left| \int_{\boldsymbol{X}} (\mathbb{E}_{\mathcal{S}}[p_{\mathcal{S}}(\boldsymbol{X})] - p_{\mathcal{S}}(\boldsymbol{X}))\mathcal{L}(f(\boldsymbol{X}), y(\boldsymbol{X}))d\boldsymbol{X} \right|}_{(II)},
\end{aligned}
$$

where $p_{\mathcal{D}}(\boldsymbol{X})$ is the density function of the distribution $\mathcal{D}$, and $p_{\mathcal{S}}(\boldsymbol{X})$ is the KDE of point $\boldsymbol{X}$.

With the smoothness of density function of $\mathcal{D}$ and Silverman (2018), we know that $(I) = O(\delta^2)$.

For (II), by using Chebychef inequality and Silverman (2018), with probability $1 - \Delta$, we have

$$(II) = O(\Delta^{-\frac{1}{2}} N^{-\frac{1}{2}} \delta^{-\frac{D}{2}} + N^{-2}).$$

On the other hand, by Taylor expansion, we know

$$\mathcal{L}_{\text{adv}}(f) - \mathcal{L}_{\text{clean}}(f) \leq O(\delta)\mathbb{E}_{(\boldsymbol{X},y)\sim\mathcal{D}}\left[ \max_{\|\boldsymbol{\xi}\|_p \leq \delta} \|\nabla_{\boldsymbol{X}} \mathcal{L}(f(\boldsymbol{X}+\boldsymbol{\xi}), y)\|_q \right].$$

Combined with the bounds for $(I)$ and $(II)$, we can derive Theorem B.1. $\qquad\square$

