# OpenReview forum: "On the Clean Generalization and Robust Overfitting in Adversarial Training from Two Theoretical Views: Representation Complexity and Training Dynamics"
_ICML.cc/2025/Conference — ICML 2025 poster_

### Official Review · Reviewer_3vVc · 2025-02-19

**Overall Recommendation:** 3

**Summary:**

I happen to review this paper again. This paper has almost no changes when compared to previous version.

This paper studies the Clean Generalization and Robust Overfitting (CGRO) of neural networks under adversarial training. It studies the CGRO from two views: representation complexity and training dynamics. A CGRO Classifier is defined as: clean test error is small, robust training error is small, but robust test error is large.

Representation complexity (Section 4): the main result shows that clean classifier requires $ploy(D)$ parameters, and CGRO requires $ploy(D + ND)$ parameters, while robust classifer requires $\exp(D)$ parameters. It shows the separation between clean/robust classifiers from approximation theory, i.e.,

Clean Classifier ($poly(D)$) ≲ CGRO Classifier($poly(D)+ND$) ≪ Robust Classifier (Ω($\exp(D)$))

In training dynamics, the results show that adversarial training makes the neural network conducts partial true feature learning and exactly memorizes spurious features. That means, after adversarial training, the network correctly classifies unseen clean data with high probability in Theorem E.12; but fails to classify the adversarial examples generated from unseen clean data with probability at least 1/2.

**Claims And Evidence:**

I have the following two concerns:

Clean Classifier ($poly(D)$) ≲ CGRO Classifier($poly(D)+ND$) ≪ Robust Classifier (Ω($\exp(D)$))

However, the results heavily depend on Assumption 4.3 rather than proven, as mentioned by two reviewers as well. Though it may observe from empirical results, it requires a careful theoretical demonstration.

Regarding the training dynamics, the statement is incomplete as the comparison to classifier under standard training is missing. In this case, it is unclear to us whether a model trained via standard (non-adversarial) training would only learn the true feature and ignore the spurious feature? Establishing this distinction is important, as it would suggest that CGRO arises from the nature of adversarial training itself rather than from the artificial construction of the data model.

**Essential References Not Discussed:**

N/A

**Experimental Designs Or Analyses:**

N/A

**Methods And Evaluation Criteria:**

This is a theoretical paper and the numerical experiments can support the theoretical findings.

**Other Comments Or Suggestions:**

N/A

**Other Strengths And Weaknesses:**

N/A

**Questions For Authors:**

N/A

**Relation To Broader Scientific Literature:**

yes, this topic is important to the machine learning community.

**Theoretical Claims:**

Yes, I checked the proof and mentioned it in the previous part on "Claims and Evidence".

---

> ### Author Rebuttal · Authors · 2025-04-01
>
> We sincerely thank the reviewer for the positive support and valuable feedback! We greatly appreciate the insightful review, and the recognition of highlighting the significance of our contribution and solidity of our theory, as well as the clarity of our writing. We are very glad to address the questions and suggestions raised by the reviewer, which we believe will help further refine our work. Below are our responses to the questions and suggestions raised by the reviewer.
>
> >**[C1]** However, the results heavily depend on Assumption 4.3 rather than proven, as mentioned by two reviewers as well. Though it may be observed from empirical results, it requires a careful theoretical demonstration.
>
> **[A1]** We sincerely thank the reviewer for the valuable and thoughtful suggestion. As the reviewer mentioned, we would like to clarify that the main theorem relies on Assumption 4.3, where we apply a teacher-student learning framework. This framework assumes the existence of a ground-truth neural network of moderate size that can achieve perfect clean classification. This teacher-student setup is widely used in deep learning theory (e.g., [1][2][3]).
>
> We also emphasize that the polynomial-size assumption arises from empirical observations, where mildly over-parameterized networks achieve good clean classification performance but poor robust classification performance (e.g., [4][5][6]), rather than from a rigorous mathematical proof. Furthermore, providing a lower bound for the representation complexity required to achieve robust generalization under Assumption 4.3 is highly non-trivial due to the complex decision boundary induced by the ground-truth polynomial-size network. To address this, we build on the technique from [7] to establish an exponential lower bound in the worst case.
>
> >**[C2]** Regarding the training dynamics, the statement is incomplete as the comparison to classifiers under standard training is missing. In this case, it is unclear to us whether a model trained via standard (non-adversarial) training would only learn the true feature and ignore the spurious feature?
>
> **[A2]** We sincerely thank the reviewer for the insightful suggestion. As the reviewer pointed out, we can prove that a model trained using standard (non-adversarial) training will only learn the true feature and ignore the spurious feature. We will include this statement in the revision of our paper.
>
> **Reference**
>
> [1] Allen-Zhu, Z., Li, Y., & Liang, Y. (2019). Learning and generalization in overparameterized neural networks, going beyond two layers. Advances in neural information processing systems, 32.
>
> [2] Lv, B., & Zhu, Z. (2022). Implicit bias of adversarial training for deep neural networks. In International Conference on Learning Representations.
>
> [3] Allen-Zhu, Z., & Li, Y. (2023, July). Backward feature correction: How deep learning performs deep (hierarchical) learning. In The Thirty Sixth Annual Conference on Learning Theory (pp. 4598-4598). PMLR.
>
> [4] Biggio, B., Corona, I., Maiorca, D., Nelson, B., Šrndi´ c, N., Laskov, P., Giacinto, G. and Roli, F. (2013). Evasion attacks against machine learning at test time. In Machine Learning and Knowledge Discovery in Databases: European Conference, ECML PKDD 2013, Prague, Czech Republic, September 23-27, 2013, Proceedings, Part III 13. Springer.
>
> [5] Szegedy, C., Zaremba, W., Sutskever, I., Bruna, J., Erhan, D., Goodfellow, I. and Fergus, R. (2013). Intriguing properties of neural networks. arXiv preprint arXiv:1312.6199.
>
> [6] Goodfellow, I. J., Shlens, J. and Szegedy, C. (2014). Explaining and harnessing adversarial examples. arXiv preprint arXiv:1412.6572.
>
> [7] Li, B., Jin, J., Zhong, H., Hopcroft, J., & Wang, L. (2022). Why robust generalization in deep learning is difficult: Perspective of expressive power. Advances in Neural Information Processing Systems, 35, 4370-4384.

---

### Official Review · Reviewer_WvLq · 2025-03-13

**Overall Recommendation:** 4

**Summary:**

This paper investigates the Clean Generalization and Robust Overfitting (CGRO) problem – defined as “robust overfitting and high clean test accuracy” (without clean overfitting/memorisation) – from perspectives of representation complexity and training dynamics. On the one hand, they show that under data assumptions of boundedness, well-separation, clean classifiers require $poly(D)$ complexity (assumption 4.3), CRGO classifiers require $poly(D) + \tilde{O}(ND)$, and adversarially robust classifiers require $exp(D)$. On the other hand, they demonstrate a three stage phase transition during learning, to understand how a convolutional classifier (on structured data) converges to robust memorisation and CRGO under adversarial training.

**Claims And Evidence:**

The first claim compares the representation complexity for clean vs. CRGO vs. robust classifiers: Clean($poly(D)$) $\lesssim$ CRGO($poly(D) + \tilde{O}(ND)$)  $\ll$ Robust($\Omega(exp(D))$). This claim relies on assumptions of bounded input, well-separated data for classification, and the existence of a ReLU clean classifier with $poly(D)$ complexity. I find these assumptions empirically reasonable for the experimental setups of CIFAR and MNIST, though I am unfamiliar with results on polynomial-sized ReLU networks for approximating clean classifiers.

The second claim concerns the dynamics of adversarial training, that the network will partially learn the true feature for well-separated classes and will exactly memorise the spurious features of specific training data. They argue that the former demonstrates clean generalisation while the latter is a case of robust overfitting (since the data-wise random noise is used for memorisation). Besides the training regime, the authors additionally connect claim 2 to CRGO in the test data regime, where the clean test error and robust training error are both small while the robust test error is significant. From what I understand, this analysis appears sound.

**Essential References Not Discussed:**

Not to my knowledge.

**Experimental Designs Or Analyses:**

To verify the complexity of CRGO and robust classifiers, the authors vary the model size and record the resultant changes in robust training loss and robust generalization gap. To examine the dynamics under adversarial training, they additionally test on synthetic, structured data, replicating the three-stage phase transition and phenomenon of CRGO. The experiments are well-aligned with the theory and provide numerical support.

**Methods And Evaluation Criteria:**

This is a predominantly theoretical work, where theoretical insights are cross-checked in practice. Experiments are conducted on the image modality (boundedness), on simple vision datasets of CIFAR-10 and MNIST (well-separated), with sufficiently expressive convolutional models of WideResNet-34 and LeNet-5, under standard $l_\infty$ PGD attacks. The clean and robust accuracies are recorded for training and unseen testing examples.

**Other Comments Or Suggestions:**

N/A

**Other Strengths And Weaknesses:**

1. **Strength (Soundness) -** This paper analyses CRGO in adversarial training from refreshing perspectives of representation complexity and training dynamics analysis. In my judgement, the analysis is technically sound and contributes to better understanding of the challenges (exponential complexity) and processes (signal and noise) underlying adversarially robust classification.
2. **Weakness (Significance) -** This paper considers adversarial training and robust classifiers under highly structured data settings. It may be challenging to determine for an arbitrary data modality, dataset and task, whether the assumptions hold and whether the theoretical insights apply.

**Questions For Authors:**

1. How would practitioners efficiently verify that a given task satisfies assumptions 4.1-4.3 in order to leverage insights from claim 1 to construct their model and vary its capacity?
2. Can claim 2 (eventual CRGO and the three stage phase transition) be demonstrated for a real, existing dataset without special construction?
3. Have the authors considered experimentally comparing against a standardly trained model baseline in Section 6.2 on the synthetically constructed dataset? This would serve as a relevant control to understand unique dynamics of adversarial training.

**Relation To Broader Scientific Literature:**

This work contrasts the representation complexity of clean vs. CRGO vs. robust classifiers, which enriches our understanding of the properties and demands of adversarial robustness. Furthermore, this work discusses the learning dynamics of networks under adversarial training, outlining the processes of partial true feature learning (which results in clean generalisation) and spurious memorisation of noisy, data-wise components (which results in robust overfitting). Together, this analysis sheds light on why CRGO eventually occurs, which is a standing problem in trying to understand adversarial training.

**Theoretical Claims:**

Referencing the above section on "claims and evidence", I examined whether the assumptions of claim 1 (4.1-4.3) are reasonable in practice; followed the proof sketch of claims 1 (section 4) and 2 (section 5) in detail; did a summary review of the claims' full proofs, comprising supplement sections D, E, F.

---

> ### Author Rebuttal · Authors · 2025-04-01
>
> We sincerely thank the reviewer for the positive feedback! We greatly appreciate the recognition of the novelty and significance of our contribution to the topic of adversarial robustness in the deep learning community, as well as the positive remarks on the clarity of our writing. We are very glad to address the questions and suggestions raised by the reviewer, which we believe will help further refine our work. Below are our responses to the questions and suggestions raised by the reviewer.
>
> >**[Q1]** How would practitioners efficiently verify that a given task satisfies assumptions 4.1-4.3 in order to leverage insights from claim 1 to construct their model and vary its capacity?
>
> **[A1]** For Assumption 4.1, due to the normalization (or centralization) of the data, we assume that Assumption 4.1 holds for general deep learning problems. For Assumption 4.2, we compute the $\ell_{p}$ distance between data from different classes in the training or test set, as done in the empirical work [1], to validate Assumption 4.2. For Assumption 4.3, we train a suitably-sized ReLU neural network (with network width equal to $\operatorname{poly}(D)$ and network depth $L$ as a constant, resulting in a total parameter count of $\operatorname{poly}(D)$) as a clean classifier. We then test its clean test accuracy and robust test accuracy to verify whether Assumption 4.3 holds. We thank the reviewer for this insightful suggestion, and we will add the above discussion in the revision of our paper.
>
> >**[Q2]** Can claim 2 (eventual CRGO and the three stage phase transition) be demonstrated for a real, existing dataset without special construction?
>
>
> **[A2]**  We would like to clarify that the patch structure we use can be seen as a simplification of real-world vision-recognition datasets. Specifically, images are divided into signal patches that are meaningful for classification, such as the whisker of a cat or the nose of a dog, and noisy patches, like the uninformative background of a photo. Our assumption about patch data can also be generalized to situations where there exists a set of meaningful patches. However, analyzing the learning process in such cases would complicate our explanation and obscure the main idea we wish to present. Therefore, we focus on the case of a single meaningful patch in our work. We would like to point out that, for real data and real models, it is difficult to rigorously define true feature learning and spurious feature learning. As a result, verifying the phase transition in real-world experiments is challenging, and this issue is commonly encountered in feature learning theory papers, such as [2][3][4][5]. We also believe that validating this on real data is an important and promising direction for future research.
>
> >**[Q3]** Have the authors considered experimentally comparing against a standardly trained model baseline in Section 6.2 on the synthetically constructed dataset? This would serve as a relevant control to understand unique dynamics of adversarial training.
>
> **[A3]** We thank the reviewer for the valuable suggestion. We have added a standardly trained model baseline on the synthetically constructed dataset.  The experiment results are presented as follows:
> |                      | Train | test |
> |----------------------|-------|------|
> | Clean Acc            | 100.0 | 100.0 |
> | Robust Acc           | \ | 1.5 |
> We will include this experiment in the revision of our paper.
>
> **Reference**
>
> [1] Yang, Y. Y., Rashtchian, C., Zhang, H., Salakhutdinov, R. R., & Chaudhuri, K. (2020). A closer look at accuracy vs. robustness. Advances in neural information processing systems, 33, 8588-8601.
>
> [2] Allen-Zhu, Z. and Li, Y. (2023b). Towards understanding ensemble, knowledge distillation and self-distillation in deep learning. In The Eleventh International Conference on Learning Representations.
>
> [3] Chidambaram, M., Wang, X., Wu, C. and Ge, R. (2023). Provably learning diverse features in multi view data with midpoint mixup. In International Conference on Machine Learning. PMLR.
>
> [4] Chen, Z., Deng, Y., Wu, Y., Gu, Q., & Li, Y. (2022). Towards understanding mixture of experts in deep learning. arXiv preprint arXiv:2208.02813.
>
> [5] Zou, D., Cao, Y., Li, Y., & Gu, Q. (2023, July). The benefits of mixup for feature learning. In International Conference on Machine Learning (pp. 43423-43479). PMLR.

---

> > ### Comment · Reviewer_WvLq · 2025-04-08
> >
> > Thank you for the detailed response. The authors have addressed my concerns regarding whether assumptions 4.1-4.3 could be satisfied in practice; the proposed sanity check for 4.3 is reasonable, as is the explanation given for A2. I find this paper to be a valuable addition to adversarial training, especially since it injects a fresh perspective (from approximation theory and feature learning) to CRGO and robust generalisation gap problems. I find it important to also articulate these insights in empirical terms and look forward to the extended discussion on how assumptions are satisfied in practice / how the theory can inform practitioners' design choices (e.g. when attempting to construct and train a robust model). To eliminate ambiguity, I raise my score from a 3 (borderline/weak accept) -> 4 (accept).

---

> > > ### Author Response · Authors · 2025-04-08
> > >
> > > We sincerely thank the reviewer for the thoughtful and encouraging feedback. We're glad that the clarifications regarding Assumptions 4.1–4.3 and the sanity check for Assumption 4.3 were helpful. We also appreciate your recognition of the theoretical perspective introduced in our work and its relevance to CGRO and the robust generalization gap.
> > >
> > > In the revision, we will make sure to elaborate further on how these assumptions are practically satisfied, and discuss how the theoretical insights can guide robust model design in real-world settings. Our goal is to make the theoretical framework not only rigorous but also actionable for practitioners. We truly appreciate your support and are encouraged by the increased score!

---

### Official Review · Reviewer_wBzb · 2025-03-14

**Overall Recommendation:** 3

**Summary:**

This study focuses on the phenomenon of clean generalization and adversarial overfitting. The authors theoretically formulate this phenomenon and analyze it from the perspective of representation complexity and learning dynamics. First, they derive the complexity required to learn CGRO models and robust models, showing that robust models require more complexity than CGRO models. Second, the authors study the learning dynamics to discover three stages in training.

**Claims And Evidence:**

Remark 4.8 states that “which may lead the classifier trained by adversarial training to the CGRO regime.” However, the theoretical analysis is not agnostic to adversarial training and it cannot explain the effects of adversarial training.

**Essential References Not Discussed:**

Liu et al., (2023) also studied the relationship between network architecture (weight sparsity) and adversarial generalization.

Liu et al., Exploring the Relationship between Architectural Design and Adversarially Robust Generalization, in CVPR 2023.

**Experimental Designs Or Analyses:**

I’m concerned about how experiments in Table 1 and Figure 2 support the theoretical analysis. I expect a comparison between models with only linear complexity and models with exponential complexity.

**Methods And Evaluation Criteria:**

- The main result of this paper, Theorem 4.4, is based on Assumption 4.3, but Assumption 4.3 is not proven or justified. Especially, the restriction of $poly(D)$ requires justification.

- Why is the function in Lemma 4.5 a CGRO model? While a CGRO classifier needs to satisfy three conditions in Definition 3.4, the function $f_S$ in Lemma 4.5 is not proven to satisfy the third one, i.e., $L^{p,\delta}_D (f) = \Omega (1)$.

- The analysis from the perspective of representation complexity fails to explain the mechanism of adversarial training. Moreover, many methods (e.g., (Wu et al., 2020)) have been proposed to improve the adversarial generalization of DNNs, and this study fails to explain the effectiveness of these studies.

Wu et al., Adversarial Weight Perturbation Helps Robust Generalization, in NeurIPS 2020.


- The analysis from the perspective of learning dynamics is limited to a specific dataset and two-layer network with a pre-defined parameter structure. Its scalability to real datasets and complex architectures is questioned.

**Other Comments Or Suggestions:**

Previous studies have discovered that adversarial robust generalization requires more data. It would be better to use the theoretical analysis in this study to explain this discovery, since the theoretical analysis here is also related to the dataset size.

**Other Strengths And Weaknesses:**

No.

**Questions For Authors:**

No.

**Relation To Broader Scientific Literature:**

This study provides two perspectives to understand the prior discovery of the adversarial overfitting phenomenon.

**Theoretical Claims:**

Assumption 4.3 is not justified.

Whether the function in Lemma 4.5 belongs to CGRO classifiers needs clarification.

---

> ### Author Rebuttal · Authors · 2025-04-01
>
> We sincerely thank the reviewer for the thoughtful feedback. We are very glad to address the questions and suggestions raised by the reviewer, which we believe will help further refine our work. Below are our responses to the questions and suggestions raised by the reviewer.
>
> **Response to claims and evidences:**
> >See our response to **[Q3-1]**.
>
> **Response to methods and evaluation criteria:**
> >**[Q1]** As the reviewer mentioned, we would like to clarify that the main theorem relies on Assumption 4.3, where we apply a teacher-student learning framework. This framework assumes the existence of a ground-truth neural network of moderate size that can achieve perfect clean classification. This teacher-student setup is widely used in deep learning theory (e.g., [1][2][3]). We also emphasize that the polynomial-size assumption arises from empirical observations, where mildly over-parameterized networks achieve good clean classification performance but poor robust classification performance (e.g., [4][5][6]), rather than from a rigorous mathematical proof. Furthermore, providing a lower bound for the representation complexity required to achieve robust generalization under Assumption 4.3 is highly non-trivial due to the complex decision boundary induced by the ground-truth polynomial-size network. To address this, we build on the technique from [7] to establish an exponential lower bound in the worst case.
>
> >**[Q2]**  We sincerely thank the reviewer for the valuable and thoughtful suggestion. Indeed, if the data distribution $\mathcal{D}$ satisfies that the covering number of the supporting set is exponential in the data input dimension $D$, we can rigorously prove the third condition, i.e., $L_{\mathcal{D}}^{p,\delta}(f) = \Omega(1)$, which complete the proof of Lemma 4.5. We will include this in the revision of our paper.
>
> >**[Q3-1]** We would like to clarify that, from the perspective of expressive power, our analysis demonstrates that CGRO classifiers can be achieved with only polynomial representation complexity (Theorem 4.4), whereas an exactly robust classifier necessitates representation complexity that is exponential in the worst case (Theorem 4.7). Given the simplicity bias inherent in neural network training (e.g., [8][9][10][11]), this fundamental complexity gap may explain the implicit bias observed in adversarial training.
>
> >**[Q3-2]** We would like to clarify that our paper focuses on the underlying mechanism behind CGRO, and explaining the effectiveness of robustness improvement methods (such as [12]) is beyond the scope of our paper.
>
> >**[Q4]** We simplify real-world vision-recognition datasets by dividing images into meaningful patches (e.g., a cat's whisker or a dog's nose) and noisy ones (e.g., an uninformative background). While this can be generalized to scenarios with multiple meaningful patches, analyzing such cases would complicate our explanation. Hence, we focus on a single meaningful patch. Defining true versus spurious feature learning in real data and models is difficult, making phase transition verification challenging, as seen in feature learning theory papers. We believe validating this on real data is a promising direction for future research.
>
> **Response to experimental designs and analyses:**
> > We would like to emphasize that our exponentially large lower bound for representation complexity holds only in the worst case. On the other hand, the computational cost of training neural networks with exponentially large input dimensions is unacceptable in practice. Therefore, as an alternative, we conducted experiments on MNIST and CIFAR10 by appropriately scaling up the network, and the phenomena observed are consistent with our theory.
>
> **Response to essential references not discussed:**
> > We thank the reviewer for pointing out the related work [17]. We will include it in the related work section in the revised version of our paper.
>
> **Response to other comments and suggestions:**
> > We thank the reviewer for the valuable suggestion. Indeed, when the training data is sufficiently large, i.e., when $N = \Omega(exp(D))$ (at which point Lemma 4.5 no longer holds, as seen in our response to **[Q2]**), the upper bound of the CGRO classifier's complexity matches the lower bound of the robust classifier's complexity. This also indicates that adversarial robustness requires more data. We will include this part in the revision of our paper.
>
> **Reference (Arxiv Index)**
>
> [1] 1811.04918
>
> [2] 2001.04413
>
> [3] 2102.06701
>
> [4] 1708.06131
>
> [5] 1312.6199
>
> [6] 1412.6572
>
> [7] 2205.13863
>
> [8] 1901.06523
>
> [9] 2201.07395
>
> [10] 2110.13905
>
> [11] 2410.10322
>
> [12] 2004.05884
>
> [13] 2012.09816
>
> [14] 2210.13512
>
> [15] 2208.02813
>
> [16] 2303.08433
>
> [17] 2209.14105

---

### Official Review · Reviewer_sZef · 2025-03-15

**Overall Recommendation:** 3

**Summary:**

The authors explained the common Clean Generalization and Robust Overfitting (CGRO) phenomenon in adversarial training from a theoretical analysis. The authors first proved that a two-layer ReLU net will achieve CGRO with small extra parameters, and the ideal robust classifier requires exponential parameters. Then, as the main contribution, the authors proved that the network has a three-stage phase during training. The network learns some “true features” in the first stage, and the noise component increases in the second stage; at last, the network learns the data-specific noise, which results in CGRO.

**Claims And Evidence:**

Yes, all the claims are supported by the theoretical analysis.

**Essential References Not Discussed:**

No

**Experimental Designs Or Analyses:**

Yes, I have checked all the experimental designs and analyses. All the experiments seem fine, however, there are some concerns:
(1) In Table 1 and Figure 2, the interval of model sizes seems to be less fully reflective of model robustness, e.g., there are no results for the acc surge process in MNIST and no surge phenomenon in CIFAR10.
(2) It is not clear about the smallest/largest/average noise memorization.

**Methods And Evaluation Criteria:**

Yes, the evaluation criteria make sense to me for all the experiments.

**Other Comments Or Suggestions:**

No

**Other Strengths And Weaknesses:**

The quality of the paper is good, presenting a reasonable motivation and methodology. The paper is generally well-written and easy to follow. The paper theoretically explains the GCOR and the learning process in adversarial learning, which makes the paper inspiring.

However, these are some concerns. For example, the intervals do not seem convincing, and there is confusion about the smallest/largest/average noise memorization. On the other hand, even though the authors explain the learning process, they do not give the solution for the GCOR which may slightly diminish the contribution.

**Questions For Authors:**

As mentioned above, I have some concerns about this paper:
(1) the intervals in Table 1 and the figure need modification. There are no results between 8 and 12 in MNIST, which makes readers wonder how exactly accuracy changed during this surge. On CIFAR-10, the robust test accuracy didn't spike with robust training accuracy as it did in MNIST. Could the authors increase the intervals (e.g., less than 1 and greater than 10) or provide an explanation?

(2) It is not clear about the smallest/largest/average noise memorization in figure 2(c).

(3) From Figure 2(c) and Phase III analysis, it seems the true feature learning would not decrease during the training, so how do the authors explain the general phenomenon that robust test accuracy will decrease after some point in the training phase?

(4) similarly, why the true feature learning would not suffer catastrophic forgetting since the “signal component is now mostly dominated by the noise component” as claimed in paper?

**Relation To Broader Scientific Literature:**

The main results of this paper focus on explaining the CGRO Phenomenon from the Representation Complexity and Learning Process on Structured Data. There are several prior works that figure out the robust generalization requires more data and larger models, which may somewhat diminish the first contribution of the paper. However, the analysis of the Learning Process seems interesting and inspiring, which may provide new insights for future research.

**Theoretical Claims:**

Yes, I have checked all the theoretical claims and their proofs, including the Representation Complexity part and Learning Process part. I didn't find any significant errors in it, but I can't guarantee that it's completely correct.

---

> ### Author Rebuttal · Authors · 2025-04-01
>
> We sincerely thank the reviewer for the encouraging and insightful feedback, and for highlighting the strength of our theoretical contributions and the clarity of our writing. We are very glad to address the questions and suggestions raised by the reviewer, which we believe will help further refine our work. Below are our responses to the questions raised by the reviewer.
>
> >**[Q1]** the intervals in Table 1 and the figure need modification.
>
> **[A1]** We thank the reviewer for the valuable suggestion.
> - For MNIST, we have added experiments with model size factors of 9, 10, and 11. The experimental results are presented in the following table.
>
> | Model Size Factor | 1     | 2     | 8     | 9    | 10    | 11    | 12    | 16   |
> |-------------------|-------|-------|-------|-------|-------|-------|-------|-------|
> | **Clean Test Acc**| 11.35 | 11.35 | 11.35 | 11.35 | 11.35 | 95.24 | 95.06 | 94.85 |
> | **Robust Test Acc**| 11.35 | 11.35 | 11.35 | 11.35 | 11.35 | 73.22 | 77.96 | 83.43 |
> | **Robust Train Acc**| 11.70 | 11.70 | 11.70 | 11.70  | 11.70  | 95.50 | 99.30 | 99.50 |
>
> We can observe that when the model size factor is less than or equal to 11, the adversarially trained LeNet model fails to learn non-trivial classifiers. However, when the model size factor is greater than or equal to 12, there is a sudden improvement in model performance. This phenomenon has also been mentioned in previous empirical work [1], and studying the theoretical mechanism behind it is an interesting future direction.
>
> - For CIFAR10, we did not observe the robust test accuracy spike phenomenon (we used the WideResNet architecture here, and a model size factor smaller than 1 results in an inability to achieve a clean classifier, even with clean training). We speculate that this is due to the inherent complexity of the CIFAR10 dataset, which is much higher than that of the MNIST dataset. We also acknowledge that theoretically analyzing the occurrence or absence of the spike phenomenon is an interesting and important direction for future research.
>
>
> >**[Q2]** It is not clear about the smallest/largest/average noise memorization in figure 2(c).
>
> **[A2]** We would like to clarify that, for our two-layer neural network and a given data point $(\boldsymbol{X}, y)$, we define the noise memorization as $\mathcal{V} := \sum_{r=1}^{m}\sum_{j \in [P] \setminus \operatorname{signal}(\boldsymbol{X})} \langle \boldsymbol{w}_r^{(T)}, y \boldsymbol{X}[j] \rangle^q$, which is mentioned in lines 291-296 (right page). Then, for the $N$-size training dataset, we obtain a total of $N$ noise memorizations $\mathcal{V}_1, \mathcal{V}_2, \dots, \mathcal{V}_N$. We define the smallest, largest, and average values of these as the smallest, largest, and average noise memorization, respectively. We thank the reviewer for pointing this out, and we will include the relevant explanation in the revision of our paper.
>
> >**[Q3,4]** From Figure 2(c) and Phase III analysis, it seems the true feature learning would not decrease during the training, so how do the authors explain the general phenomenon that robust test accuracy will decrease after some point in the training phase? Similarly, why does the true feature learning not suffer catastrophic forgetting since the “signal component is now mostly dominated by the noise component” as claimed in the paper?
>
> **[A3,4]** As the reviewer mentioned, we would like to clarify that during training, the true feature learning does not decrease. However, its increment is dominated by the noise component (as seen in Lemma 5.12). Thus, our theory cannot directly explain the observation that robust test accuracy decreases after some point in the training phase and that true feature learning suffers catastrophic forgetting, which implies a gap between theory and observation. We believe that theoretically explaining this gap is an interesting and important future direction.
>
> **Reference**
>
> [1] Madry, A., Makelov, A., Schmidt, L., Tsipras, D., & Vladu, A. (2017). Towards deep learning models resistant to adversarial attacks. arXiv preprint arXiv:1706.06083.

---

### Decision · Program_Chairs · 2025-05-01

**Decision:**

Accept (poster)

**Comment:**

The paper studies the Clean Generalization and Robust Overfitting (CGRO) phenomenon from two views: representation complexity and training dynamics.

The reviewers agreed upon positive ratings. Overall I recommend accept.